# Lacticaseibacillus rhamnosus Probio-M9 extends the lifespan of Caenorhabditis elegans

Juntao Zhang[1,2,3], Yanmei Zhao[4], Zhihong Sun[1,2,3] & Tiansong Sun [1,2,3 ✉]

Probiotics have been characterized as useful for maintaining the balance of host gut flora and conferring health effects, but few studies have focused on their potential for delaying aging in the host. Here we show that *Lacticaseibacillus rhamnosus* Probio-M9 (Probio-M9), a healthy breast milk probiotic, enhances the locomotor ability and slows the decline in muscle function of the model organism *Caenorhabditis elegans*. Live Probio-M9 significantly extends the lifespan of *C. elegans* in a dietary restriction-independent manner. By screening various aging-related mutants of *C. elegans*, we find that Probio-M9 extends lifespan via p38 cascade and *daf-2* signaling pathways, independent on *daf-16* but dependent on *skn-1*. Probio-M9 protects and repairs damaged mitochondria by activating mitochondrial unfolded protein response. The significant increase of amino acids, sphingolipid, galactose and fatty acids in bacterial metabolites might be involved in extending the lifespan of *C. elegans*. We reveal that Probio-M9 as a dietary supplementation had the potential to delay aging in *C. elegans* and also provide new methods and insights for further analyzing probiotics in improving host health and delaying the occurrence of age-related chronic diseases.

[1] Inner Mongolia Key Laboratory of Dairy Biotechnology and Engineering, Inner Mongolia Agricultural University, Hohhot, Inner Mongolia, China. [2] Key Laboratory of Dairy Products Processing, Ministry of Agriculture and Rural Affairs, Inner Mongolia Agricultural University, Hohhot, Inner Mongolia, China. [3] Key Laboratory of Dairy Biotechnology and Engineering, Ministry of Education, Inner Mongolia Agricultural University, Hohhot, Inner Mongolia, China. [4] Key Laboratory of RNA Biology, CAS Center for Excellence in Biomacromolecules, Institute of Biophysics, Chinese Academy of Sciences, Beijing, China. ✉email: sts9940@sina.com

The morbidity and mortality of cardiovascular diseases, neurodegenerative diseases, cancer and other illnesses, are significantly and positively correlated with age[1]. Hence, efforts to extend the lifespan and delay the aging process have received wide attention. Many studies have focused on the key effects of specific diets and nutrient resources in anti-aging, particularly probiotics supplementation[2–5]. Living microorganism probiotics have functions that are beneficial to human health when consumed in sufficient amounts[6]. It has been demonstrated that probiotics confer health-promoting benefits to their host by improving the intestinal microbial balance[7–9], enhancing immune modulation[10,11], and/or competing with pathogens[12]. However, the mechanisms by which probiotics extend host lifespan remain largely elusive.

The nematode *Caenorhabditis elegans* is a widely used and successful model organism in anti-aging studies because of its obvious advantages, including its short and easily monitored lifespan[13,14], and the absence of ethical issues[15]. Genes that modulate aging processes in *C. elegans* include some of the classical and conserved signaling pathways also found in vertebrates. The p38 mitogen-activated protein kinase (MAPK) pathway is one of the most ancient evolutionarily conserved pathways in *C. elegans*[16]. In human, the p38 cascade is typically activated by inflammatory cytokines and pathogen invasion[17,18]. In *C. elegans*, the p38 cascade is activated by NSY-1 MAPK kinase, SEK-1 MAPK kinase and PMK-1[15,16]. Some conserved components of the insulin/insulin-like growth factor 1 (IGF-1) signaling (IIS) pathway also regulate aging. The sole insulin/IGF-1 receptor encoded by the gerontogene *daf-2*[19] is a key upstream component of IIS, regulating various physiological processes, such as aging and adult lifespan in *C. elegans*[20,21]. The lifespan extending effects of *daf-2* are mediated via the *daf-16*/FOXO transcription factor in reduced insulin signaling mutants such as *daf-2 (e1370)*[20]. SKN-1, a stress-responsive nuclear transcription factor, contributes to the reduction of IIS-associated longevity[22]. Additionally, certain substances secreted by bacterial metabolites, such as colanic acid, agmatine and methylglyoxal have been identified as key factors that have great impacts on longevity in their hosts[2,23,24].

Mitochondrial function has a far-reaching effect on the process of aging[25]. It has been shown that mitochondrial dysfunction has beneficial effects on the lifespan of *C. elegans*[26,27]. Mutations that cause electron transport chain (ETC) dysfunction extend the worm lifespan by as much as 50%[27]. The mitochondrial unfolded protein response (UPR$^{mt}$) is also associated with health-promoting and mitochondrial homeostasis[28]. UPR$^{mt}$ is activated when the transcription factor ATFS-1 shuttles from mitochondria to the nucleus in response to mitochondrial stress[29].

In this study, we explored the effects of the probiotic *Lacticaseibacillus rhamnosus* Probio-M9, which is isolated from healthy breast milk, on physiological functions including lifespan, locomotor ability and lipofuscin accumulation in *C. elegans*. Feeding conditioned with Probio-M9 significantly extended the worm lifespan. Using various aging-related mutants of *C. elegans* involving signal transduction, including mutations in the p38 MAPK, nutrient-sensing signaling pathways and UPR$^{mt}$, we investigated the molecular mechanism of lifespan extension by Probio-M9. Combined with the differential expression of metabolites by Probio-M9, our study revealed the possible metabolic pathway of Probio-M9 to delay host aging. These data reveal the vital link between probiotics and host longevity, provide new insights for the continued development of probiotics, and indicate that dietary supplementation with probiotics might have the potential to delay host aging.

## Results

**Probio-M9 extends the lifespan of *C. elegans*.** Probio-M9 and *E. coli* OP50 belong to different bacterial genera, it has been reported that nematodes exhibited a preference when the normal food *Escherichia*

*coli* OP50 (*E. coli* OP50) is replaced with other bacteria[30,31]. Therefore, Probio-M9 precipitate was resuspended in *E. coli* OP50 solution to exclude the preference of worms. In this study, we used an approximately 1:4 ratio of *E. coli* OP50 to Probio-M9 (viable plate count: $6.8 \times 10^8$ CFU/mL and $2.82 \times 10^9$ CFU/mL, respectively) as the experimental group (OP50 + Probio-M9), and *E. coli* OP50 alone as the control group (OP50). First, we used choice assays to explore whether the worms had a preference for OP50 over OP50 + Probio-M9. Worm eggs were cultured to late L4 stage on OP50 (Supplementary Fig. 1a) or OP50 + Probio-M9 (Supplementary Fig. 1c), then observed following transfer to OP50 or OP50 + Probio-M9 for 1 h or 2 h. The results showed that the number of worms on each plate was similar (Supplementary Fig. 1b, d), suggesting that *C. elegans* did not display a preference between OP50 and OP50 + Probio-M9. We also performed a binary selection analysis. Worm eggs were cultured to late L4 stage on OP50 (Supplementary Fig. 1e) or OP50 + Probio-M9 (Supplementary Fig. 1g), then transferred to plate with two bacterial lawns with OP50 and OP50 + Probio-M9. Similarly, there was no significant difference in the number of worms in each lawn after 1 h or 2 h feeding (Supplementary Fig. 1f, h). These results indicated that the worms did not exhibit any avoidance behavior or preference regarding OP50 + Probio-M9. Taken together, these results indicated that Probio-M9 is a suitable dietary supplementation for *C. elegans*.

Next, we evaluated the effect of Probio-M9 on the lifespan of *C. elegans*. Synchronized late L4 stage worms were shifted from OP50 to OP50 + Probio-M9 or continued to culture on OP50, and survival was measured daily until all of the worms had died. The results showed that OP50 + Probio-M9 significantly extended worm lifespan by about 30% (Fig. 1a and Supplementary Table 1). When the proportion of Probio-M9 in the bacterial mixture was changed (either diluted or increased), extension of lifespan was still observed (Fig. 1b and Supplementary Table 1), but was optimal in the 1:4 ratio of OP50: Probio-M9 (OP50 + Probio-M9), suggesting that the beneficial effect of Probio-M9 is dose dependent.

**Probio-M9 exhibits health-promoting effects in *C. elegans*.** Dietary restriction (DR) is well known for delaying the development of *C. elegans*, thereby prolonging its lifespan[32]. To investigate the effects of Probio-M9 feeding on the development of *C. elegans*, we examined life cycle, brood size and body size. Culturing on OP50 + Probio-M9 did not affect the time required to reach reproductive age, nor did it affect the reproductive capacity, or cycle of the worms (Fig. 2a–c). Body size of worms was also unaffected by OP50 + Probio-M9 (Fig. 2d). In addition, we investigated the effect of DR signaling pathway in lifespan extension of *C. elegans*. We found that OP50 + Probio-M9 still extended the lifespans of *eat-2 (ad1116)* and *aak-2 (ok524)* mutant worms (Fig. 2e, f and Supplementary Table 2). Together, these data showed that feeding worms with Probio-M9 had no effect on their growth and development, indicating that Probio-M9 extends the lifespan of *C. elegans* in a DR-independent manner.

In *C. elegans*, the decline in muscle function and accumulation of lipofuscin are closely correlated with aging. To investigate whether feeding with Probio-M9 might improve these physiological functions, we measured three classical phenotypes associated with aging. The pharyngeal pumping rate declined progressively with aging in both groups of worms, and feeding with OP50 + Probio-M9 had no significant difference in pharyngeal pumping rate on days 2 and 4 of young adulthood, but significantly delayed the decrease in pharyngeal pumping rate on days 6–14 of adulthood (Fig. 2g). Consistent with the changes in pharyngeal pumping rate, the decline in normal locomotor ability of the

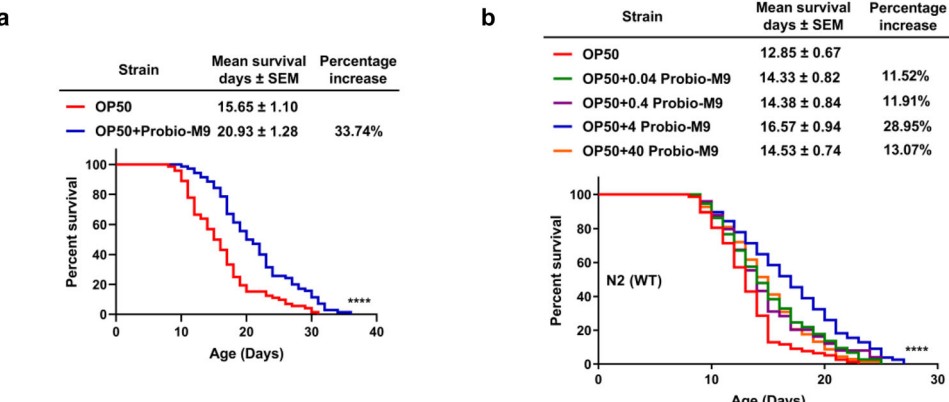

**Fig. 1 Effects of Probio-M9 feeding on regulation of lifespan in *C. elegans*. a** Probio-M9 precipitate was resuspended in OP50 significantly extended the lifespan of wild-type N2 worms ($p < 0.0001$, Log rank test). **b** The most suitable dose of Probio-M9 (OP50 + 4 Probio-M9) significantly extended the lifespan of wild-type N2 worms ($p < 0.0001$, Log rank test). The lower (OP50 + 0.04 Probio-M9 and OP50 + 0.4 Probio-M9) or higher (OP50 + 40 Probio-M9) amount of Probio-M9 exhibits less extension on lifespan ($p < 0.05$, Log rank test). In **a** and **b**, lifespans were determined at least in three biologically independent experiments, and 90 worms of OP50 or OP50 + Probio-M9 feeding were tested in the single experiment, result from a single representative experiment is shown. The amount of OP50 and Probio-M9 which were used is $6.8 \times 10^8$ CFU/mL and $2.82 \times 10^9$ CFU/mL, respectively.

worms was also delayed by OP50 + Probio-M9 feeding (Fig. 2h). Furthermore, we found that the accumulation of lipofuscin, an autofluorescent compound that accumulates in aging cells, was significantly reduced in worms fed with OP50 + Probio-M9 compared with those fed with OP50 on days 10 and 15 of adulthood (Fig. 2i, j). These findings indicated that Probio-M9 feeding supports the maintenance of higher muscle mass and enhances physical function in older worms.

Because longevity and stress resistance are interrelated, we also examined the effect of Probio-M9 on resistance to physical (temperature) stress in *C. elegans*. At day 5 of adulthood, worms fed with OP50 + Probio-M9 displayed significantly higher survival rates, than those fed with OP50, following 18 or 24 h of exposure to 34 °C heat (Fig. 2k). These results suggested that Probio-M9 has a positive effect on the thermostability of *C. elegans*. Altogether, our data indicated that Probio-M9 feeding improves the health of worms and extends their lifespan.

**The health-promoting effects of Probio-M9 depends on its adhesion to the host gut**. Several different standards have been used to screen potential bacterial strains for probiotics usefulness. The adhesion of probiotics to intestinal epithelial cells, and the intestinal accumulation of probiotics are vital prerequisites for maintaining homeostasis in the host gut microbiota[33]. To evaluate the effects of Probio-M9 on the intestinal health of the *C. elegans*, we examined the effects of feeding intervention with OP50 + Probio-M9 at specific developmental stages on the lifespan of worms, and tested the adhesion of Probio-M9 in the worm intestine. Worms derived from eggs that were cultured to either the L4 stage or adulthood (day 2 or day 4 after L4) on OP50 + Probio-M9 exhibited a significantly longer lifespan extension compared with those fed with OP50 (Fig. 3a–e and Supplementary Table 3). We also found that worms cultured from eggs to adulthood (day 4 after L4) on OP50 + Probio-M9 had a significantly longer lifespan extension than those cultured from eggs to the L4 stage on OP50 + Probio-M9 (Fig. 3f and Supplementary Table 3). These results indicated that the extent of lifespan extension is dependent on timing of on Probio-M9 feeding in the developmental cycle of *C. elegans*.

To verify the presence of Probio-M9 in the intestine of *C. elegans*, we assessed the ability of Probio-M9 to adhere to the worm intestinal tract on days 5 and 10 of adulthood. After exposure to OP50 + Probio-M9 for 5 or 10 days, the total number

of colonies forming units (CFU) was used to determine the number of bacteria accumulated in the intestine. The number of Probio-M9 on days 5 and 10 (217 CFU/mL/worm and 793 CFU/mL/worm, respectively), were comparable to those in the positive control *Lacticaseibacillus rhamnosus* GG (LGG, 166 CFU/mL/worm and 872 CFU/mL/worm, respectively) (Fig. 3g). To confirm that Probio-M9 remained in the intestine, day 10 worms fed with OP50 + Probio-M9 or OP50 + LGG were transferred to OP50 for 3 days. The total number of Probio-M9 colonies was 667 CFU/mL/worm in those previously fed with OP50 + Probio-M9, similar to the 747 CFU/mL/worm in the LGG group (Fig. 3h). These results demonstrated that Probio-M9 is maintained in the intestinal tract of *C. elegans* and contributes to the healthspan of the host.

In addition, we examined the impact of feeding with heat-inactivated Probio-M9 (95 °C for 30 min) on the lifespan of *C. elegans*. We found that OP50 + heat-inactivated Probio-M9 was unable to extend the lifespan of worms (Fig. 3i and Supplementary Table 3), suggesting that Probio-M9 must be in the live state to exert beneficial effects on the host's healthspan. Taken together, these results indicated that the health-promoting effects of Probio-M9 involve adherence to the intestinal tract of *C. elegans*.

**Probio-M9 acts through the p38 MAPK signaling pathway**. To understand the molecular mechanisms of longevity associated with Probio-M9 feeding, we explored the p38 signaling pathway, a conserved MAPK subfamily signaling pathway that plays a pivotal role in host lifespan regulation[16]. Using *C. elegans* mutants, we first investigated whether NSY-1, SEK-1 and PMK-1, three key factors in the p38 signaling pathway, were associated with host longevity induced by Probio-M9 feeding. Interestingly, we found that *nsy-1* (*ag3*) and *pmk-1* (*km25*) mutation almost completely suppressed the lifespan extension associated with Probio-M9 feeding, *sek-1* (*km4*) mutation strongly but incompletely suppressed the longevity (~30% lifespan extension in N2 vs ~12% lifespan extension in the *sek-1* mutant) (Fig. 4a–c and Supplementary Table 4), suggesting that the *C. elegans* p38 cascade is involved in the lifespan extension mediated by Probio-M9.

To further verify the involvement of the p38 signaling pathway, we evaluated the effect of Probio-M9 on the lifespan of two mutants of the Toll-interleukin-1 resistance (TIR-1) domain protein *tir-1* (*tm3036*) and *tir-1* (*ok1052*), which functions upstream of PMK-1 in *C. elegans*. Similarly, OP50 + Probio-M9 was unable to extend

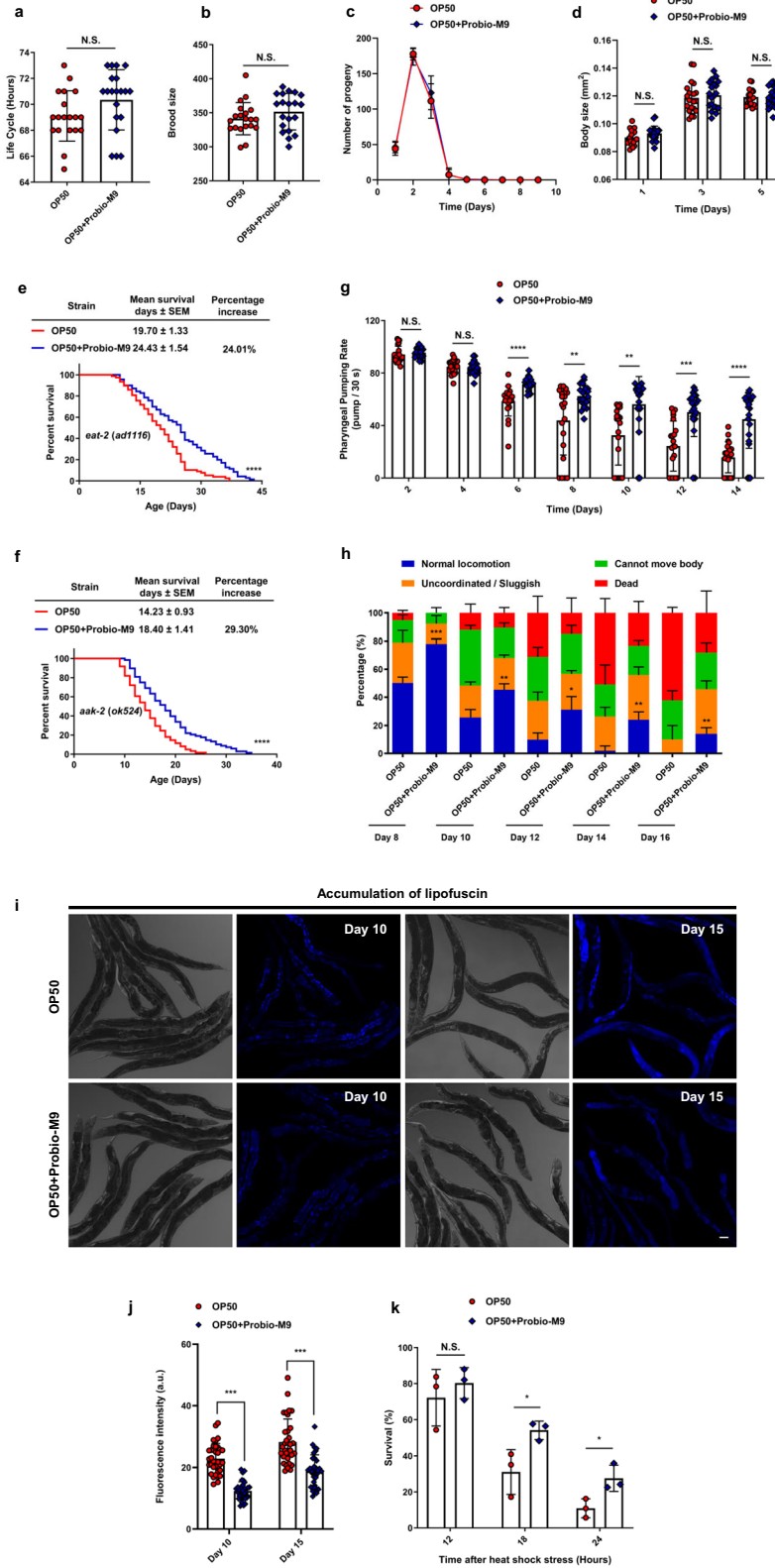

the lifespan of either mutant (Fig. 4d, e and Supplementary Table 4). As further confirmation, we explored the effect of Probio-M9 on ATF-7, a transcription factor that acts downstream of PMK-1 in the p38 signaling pathway. Again, OP50 + Probio-M9 was unable to extend the lifespan of atf-7 (gk715) mutant worms (Fig. 4f and Supplementary Table 4), suggesting that Probio-M9 might activate ATF-7 through PMK-1 to extend the lifespan of C. elegans.

Altogether, our data indicated that the effects of Probio-M9 on lifespan extension in C. elegans require the p38 signaling pathway.

**Probio-M9 downregulates the insulin-like signaling pathway.** The interactions between host and probiotic in the promotion of longevity are complex and multifactorial[34]. We wondered whether Probio-M9 might work through one of the well-known

**Fig. 2 Probio-M9 exhibits health-promoting effects.** The life cycle (from egg to its first egg laid) (**a**), total number of progeny (**b**), reproductive cycle (**c**) and body size (**d**) from wild-type N2 worms grown on OP50 and OP50 + Probio-M9 are not significantly different ($N = 20$ worms, $p > 0.05$, Student's $t$ test). Survival curves of mutants of DR signaling pathways, eat-2 (ad1116) (**e**) and aak-2 (ok524) (**f**), fed with Probio-M9 ($N = 90$ worms, $p < 0.0001$, Log rank test). **g** The pharyngeal pumping rate from wild-type N2 worms grown on OP50 + Probio-M9 was enhanced on days 2–14 compared with those fed with OP50 ($N = 20$ worms, $p < 0.01$, two-way ANOVA). **h** The normal locomotor activity from wild-type N2 worms grown on OP50 and OP50 + Probio-M9 was measured on days 8–16. According to their locomotion, worms were divided into four classes: Class A, normal locomotion, spontaneous and/or rhythmical sinusoidal movement (blue bars); Class B, cannot move body, irregular and/or uncoordinated movement (orange bars); Class C, uncoordinated/sluggish, moved merely their head and/or tail in respond to a moderate touch with a platinum wire picker (green bars); and Class D, dead, dead worms (red bars) ($N \geq 90$ worms, $p < 0.05$, Chi-squared test). **i, j** The lipofuscin accumulation of wild-type N2 worms was weakened followed OP50 + Probio-M9 feeding. Accumulation of lipofuscin was detected by autofluorescence under FV1200 confocal microscope on day 10 or 15 worms, scale bar, 20 μm (**i**), and fluorescence intensity was measured using ImageJ software (**j**). Results are shown by arbitrary units (a.u.) ($N = 34$ worms, $p < 0.001$, Student's $t$ test). **k** The thermotolerance of wild-type N2 worms grown on OP50 + Probio-M9 increased the survival rate compared those feds with OP50. Day 5 of adulthood worms cultured on OP50 or OP50 + Probio-M9 were transferred from 20 °C to 34 °C, and the survival rate was counted at 12 h, 18 h and 24 h, respectively ($N = 3$ biologically experiments, $p < 0.05$, Student's $t$ test). In **a–d**, **g**, **h**, **j** and **k**, values are presented as the mean ± SEM.

nutrient-sensing mechanisms in the host. Using *C. elegans* with loss-of-function mutants in the IIS and target of rapamycin (TOR) signaling pathways, we investigated whether these host factors were associated with host longevity through Probio-M9 feeding. Of the four mutants tested, only the daf-2 (e1370) mutant failed to exhibit an extended lifespan in response to feeding with OP50 + Probio-M9 (Fig. 5a, b, Supplementary Fig. 2a, b and Supplementary Tables 5, 6). This result suggested that the mechanism of Probio-M9 lifespan extension involves the IIS signaling pathway, but not the TOR signaling pathway in *C. elegans*.

DAF-2 is an upstream component of IIS and regulates a variety of physiological processes[35]. To determine how Probio-M9 interacts with the IIS signaling pathway in the regulation of longevity, we examined the potential involvement of DAF-2 downstream transcription factors HSF-1 (heat shock transcription factor 1), SKN-1/nuclear factor erythroid 2 (NRF2) and DAF-16/FOXO in lifespan extension. We found that Probio-M9-mediated extension of the *C. elegans* lifespan was independent of DAF-16, but dependent on SKN-1 and HSF-1 (Fig. 5c–e and Supplementary Table 5), suggesting that SKN-1 and HSF-1 might be required for the longevity effect of Probio-M9.

SKN-1 and HSF-1 function as stress resistance transcription factors[22,36]. To confirm whether Probio-M9 feeding improves stress resistance and extends the lifespan of *C. elegans*, we investigated two GFP reporters skn-1::gfp (transcriptional target of skn-1) and hsp-16.2p::gfp (a direct transcriptional target of hsf-1) to reflect the stress resistance SKN-1 and HSF-1, respectively. Our results showed that the expression of skn-1::gfp increased in the intestine of worms fed with OP50 + Probio-M9, but had no effect on hsp-16.2p::gfp (Fig. 5f–i). Furthermore, we detected the expression of GST-4, an indicator of SKN-1 activity[37], and the results indicated that the expression of gst-4p::gfp increased in *C. elegans* fed with OP50 + Probio-M9, compared with those fed with OP50 (Fig. 5j, k). Taken together, these results suggested that Probio-M9 influences lifespan extension and stress resistance by a skn-1-dependent but hsf-1-independent mechanism.

**Probio-M9 acts on host mitochondria to promote longevity.** To further probe the molecular mechanism underlying the stress resistance effect of Probio-M9, we investigated two other organelle-specific stress responses endoplasmic reticulum unfolded protein response (UPR^ER) and UPR^mt. Probio-M9 did not affect the induction of GFP reporter of stress gene in the UPR^ER (hsp-4p::gfp) (Supplementary Fig. 3a, b), it did have an impact on mitochondrial stress. Particularly, the expression of hsp-6p::gfp increased in the intestine and tail of worms fed with OP50 + Probio-M9 (Fig. 6a, b), suggesting that feeding with Probio-M9 induces the UPR^mt.

To confirm the involvement of molecular components known to activate the UPR^mt, we first checked ATFS-1, a pivotal transcription

factor in UPR^mt regulation. As expected, OP50 + Probio-M9 was unable to extend the lifespan of the loss-of-function atfs-1 (gk3094) mutant (Fig. 6c and Supplementary Table 7). We also examined the effects of Probio-M9 on loss-of-function mutants isp-1 (qm150) and nuo-6 (qm200) to evaluate whether ISP-1 and NUO-6, components of the ETC, were associated with Probio-M9-mediated promotion of host longevity. Again, OP50 + Probio-M9 was unable to extend the lifespan of the isp-1 (qm150) or nuo-6 (qm200) mutants (Fig. 6d, e and Supplementary Table 7), suggesting that the lifespan extending effects of Probio-M9 converge with the pro-longevity effects of reduced mitochondrial function in *C. elegans*. These results showed that Probio-M9 extends its host's lifespan by enhancing the UPR^mt and thus maintaining mitochondrial homeostasis in intestinal cells.

To evaluate the effect of Probio-M9 on mitochondrial membrane potential, we used a mitochondria-specific fluorescent dye, tetramethylrhodamine ethyl ester (TMRE), to stain worms. The TMRE fluorescence intensity was significantly increased in the pharyngeal bulbs of worms fed with OP50 + Probio-M9 compared with OP50 (Fig. 6f, g), suggesting that Probio-M9 increases the mitochondrial membrane potential of the host. Because the membrane potential across the inner membrane is the major driving force for mitochondria to produce ATP, Probio-M9 feeding might contribute to ATP production and storage in the cells and body of the *C. elegans*.

**Identification and analysis of differential metabolites between OP50 and Probio-M9.** To elucidate the mechanism underlying Probio-M9-mediated longevity, we performed metabolomics analysis using ultra-performance liquid chromatography-mass spectrometry to identify differentially regulated metabolites between OP50 and Probio-M9. Total ion chromatography (TIC) and principal component analysis (PCA) were used to analyze and evaluate the quality control (QC) bacterial data collected in the experiment. The TIC for samples was tested in positive and negative modes. These analyses revealed a high level of superposition when comparing the retention time and intensity across each chromatographic peak, reflecting high instrument stability (Supplementary Fig. 4a, b). PCA scores showed that the mixed quality control bacteria (mix01, mix02 and mix03) were well-gathered, but were separated from OP50 and OP50 + Probio-M9, indicating that the repeatability of the experimental conditions and data obtained were stable and reliable (Supplementary Fig. 4c).

Totally 1,253 metabolites were identified and quantified. PCA, orthogonal partial least squares discriminant analysis (OPLS-DA) models, and a heat map were used to evaluate the presence of metabolic variations between OP50 and OP50 + Probio-M9. PCA scores, OPLS-DA score plots and the heat map indicated that OP50 and OP50 + Probio-M9 were distributed in different regions,

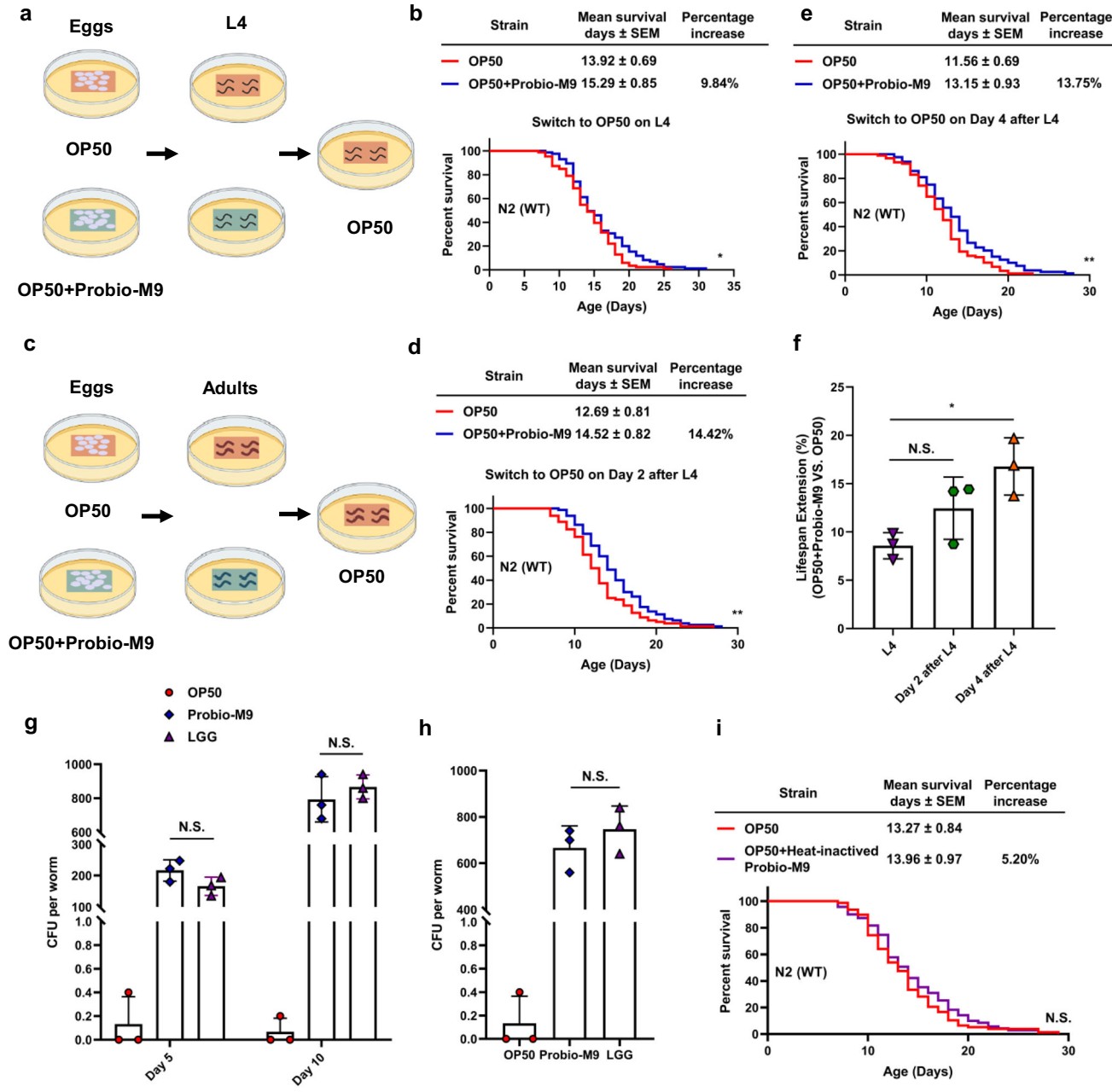

**Fig. 3 Viability and host accumulation capacity of Probio-M9. a, c** Schematics illustrate the method and bacteria used in each assay. After bleaching, eggs were transferred to bacterial lawn (OP50 or OP50 + Probio-M9) and allowed to develop to late L4 stage or adulthood (day 2 or day 4 after L4 stage). **b, d, e** Survival curves of OP50 or OP50 + Probio-M9-fed to late L4 stage (**b**), day 2 after L4 stage (**d**) and day 4 after L4 stage (**e**) (N = 90 worms, p < 0.05, Log rank test). **f** OP50 + Probio-M9 feeding at day 4 after L4 stage significantly lengthened lifespan, but not at day 2 after L4 stage, compared with those fed with OP50 + Probio-M9 to L4 stage (N = 3 biologically experiments, p < 0.05, Student's t test). **g** Colony forming indicated that Probio-M9 was attached to the *C. elegans* gut similar to LGG (N = 3 biologically experiments, p > 0.05, Student's t test). Day 5 or day 10 worms were collected from the NGM plates lawned with the test bacterial and processed to count the number of bacterial CFU, expressed as CFU per worm. OP50 and LGG were used as control. **h** Accumulation of the *C. elegans* gut by Probio-M9. Worms were fed with Probio-M9 or control (OP50 and LGG) for 10 days, and then transferred them to new plates lawn with OP50. After 3 days, the number of CFU in the worm gut was calculated, expressed as CFU per worm, and the results indicated that Probio-M9 was still attached to *C. elegans* gut (N = 3 biologically experiments, p > 0.05, Student's t test). In **f–h**, values are presented as the mean ± SEM. **i** Heat-inactivated Probio-M9 (95 °C for 30 min) failed to extend the lifespan of wild-type N2 worms (N = 90 worms, p > 0.05, Log rank test).

without overlap or crossover, instead, with obvious separation (Fig. 7a–c). Differentially regulated metabolites between OP50 and OP50 + Probio-M9 were considered to be statistically significant when the change in metabolite level simultaneously met the criteria of a fold change (FC) ≤ 0.5 or ≥ 2 and a variable importance in projection (VIP) score threshold ≥ 1. These metabolites were

then represented using a volcano plot (Fig. 7d), with the details summarized in Supplementary Data 1. Metabolites showing significant differences were divided into 14 main classes, including amino acids and their metabolites, carboxylic acids and its derivatives, organic acids and its derivatives, sphingolipid, nucleotides and their metabolites, benzene and its derivatives, and

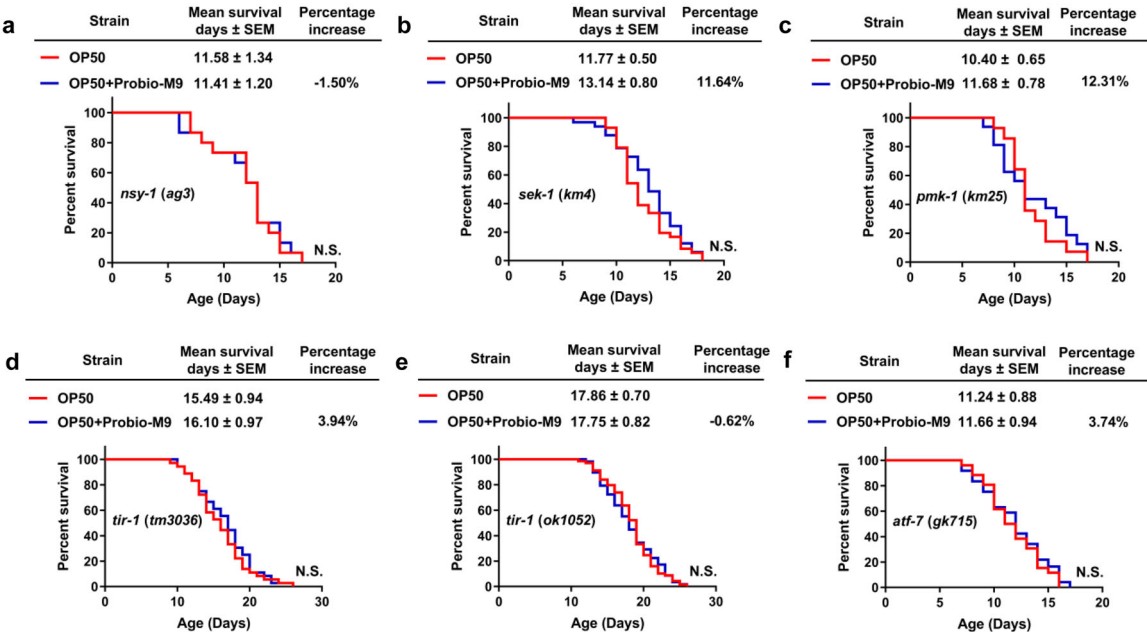

**Fig. 4 Probio-M9 acts on p38 MAPK signaling pathway to delay aging.** Survival curves of p38 MAPK signaling pathway mutants, *nsy-1* (*ag3*) (**a**), *sek-1* (*km4*) (**b**), *pmk-1* (*km25*) (**c**), *tir-1* (*tm3036*) (**d**), *tir-1* (*ok1052*) (**e**) and *atf-7* (*gk715*) (**f**) followed Probio-M9 feeding (N = 90 worms, p > 0.05, Log rank test).

heterocyclic compounds. The top ten up-regulated and top ten down-regulated metabolites with Log2FC values between OP50 and OP50 + Probio-M9 are shown in Fig. 7e, reflecting the levels of variation in metabolites in these pathways. To correlate the Probio-M9-associated changes with metabolic pathways, the Kyoto Encyclopedia of Genes and Genomes (KEGG) pathway database was used to identify pathway enrichment patterns for metabolites with significant differences. Eighty-two metabolic pathways were identified as being affected by the metabolic alterations, (Supplementary Fig. 4d), with twenty of the significantly changed metabolites contributing to enrichment in pathways related to tryptophan metabolism, fatty acid biosynthesis, alanine, aspartate and glutamate metabolism, ferroptosis, galactose metabolism and sphingolipid metabolism (Fig. 7f), indicating that these pathways might contribute to the host lifespan extension mediated by Probio-M9.

## Discussion

This study revealed that feeding with *Lacticaseibacillus rhamnosus* Probio-M9, which is isolated from healthy breast milk, can extend the lifespan of the model organism *C. elegans* (Fig. 8). We found that feeding with live Probio-M9, but not heat-inactivated Probio-M9, improved the health of worms. Probio-M9 extended the lifespan of *C. elegans* by regulating the p38 cascade and *daf-2* signaling pathways, independent on *daf-16* but dependent on *skn-1*. By activating the UPR$^{mt}$, it also appears to protect and repair damaged mitochondria to maintain metabolic stability. Meanwhile, amino acid metabolism, sphingolipid metabolism, galactose metabolism and fatty acid metabolism may be involved in the Probio-M9-mediated longevity of *C. elegans*.

It has been reported that probiotics benefit their hosts by improving the intestinal microecological balance[38–40], regulating the immune system[10,41,42] and attaching to the intestinal tract[43]. We previously reported that Probio-M9 has the ability to survive in high bile salts or at a low pH in vitro[44]. Additionally, we found that Probio-M9 inhibits the formation of colorectal tumors in mice by regulating the intestinal environment and improving the inflammatory response[45].

DR has been shown to extend lifespan and delay aging in a variety of species[32,46,47]. Although DR is known to extend the lifespan of *C. elegans*, the worms under DR are smaller in size than those fed with a normal diet[48]. In this study, the body size and development of worms fed with OP50 + Probio-M9 were normal, and the developmental cycle was similar to those fed with OP50. Furthermore, OP50 + Probio-M9 feeding extended the lifespan of loss-of-function *eat-2* (*ad1116*) and *aak-2* (*ok524*) mutants of the DR signaling pathway, indicating that Probio-M9 extends the lifespan of *C. elegans* in a DR-independent manner. Consistent with these findings, *Lactobacillus gasseri* SBT2055 was shown to extend the lifespan of *C. elegans* by 37% in a DR-independent manner[49].

Decreased muscle function and lipofuscin accumulation are closely related to aging[2,50,51]. Our study showed that feeding with Probio-M9 enhanced locomotor ability and reduced lipofuscin accumulation in older worms, indicating that Probio-M9 improves the healthspan of the host and extends their lifespan. This is consistent with a previous report that *Weissella* also increased locomotor ability and lowered the accumulation of lipofuscin in older worms[52]. Furthermore, at day 5 of adulthood, the survival rate of worms exposed to high temperature (34 °C) was obviously higher in those fed with OP50 + Probio-M9 than those fed with OP50, demonstrating that Probio-M9 feeding confers worms with an increased ability to endure the environmental stress. Consistent with our findings, *Bacillus subtilis* NCIB3610 was shown to extend the lifespan of *C. elegans* by enhancing resistance to heat stress, osmotic stress, metal stress and $H_2O_2$ oxidative stress[53]. The present study, which demonstrates that Probio-M9 feeding can enhance the stress response and extend the lifespan of *C. elegans*, provides a theoretical basis for further study of Probio-M9 in delaying aging processes in its host.

Because the intestinal cells of *C. elegans* are structurally similar to those of humans, they are considered an ideal tool to explore bacteria-host communications in the intestines[54]. The ability to adhere to the intestinal lining, a prerequisite for the colonization and physiological or pathophysiological effects of both detrimental and salutary bacteria in the body, is a key indicator for screening probiotic strains and reducing the accumulation of

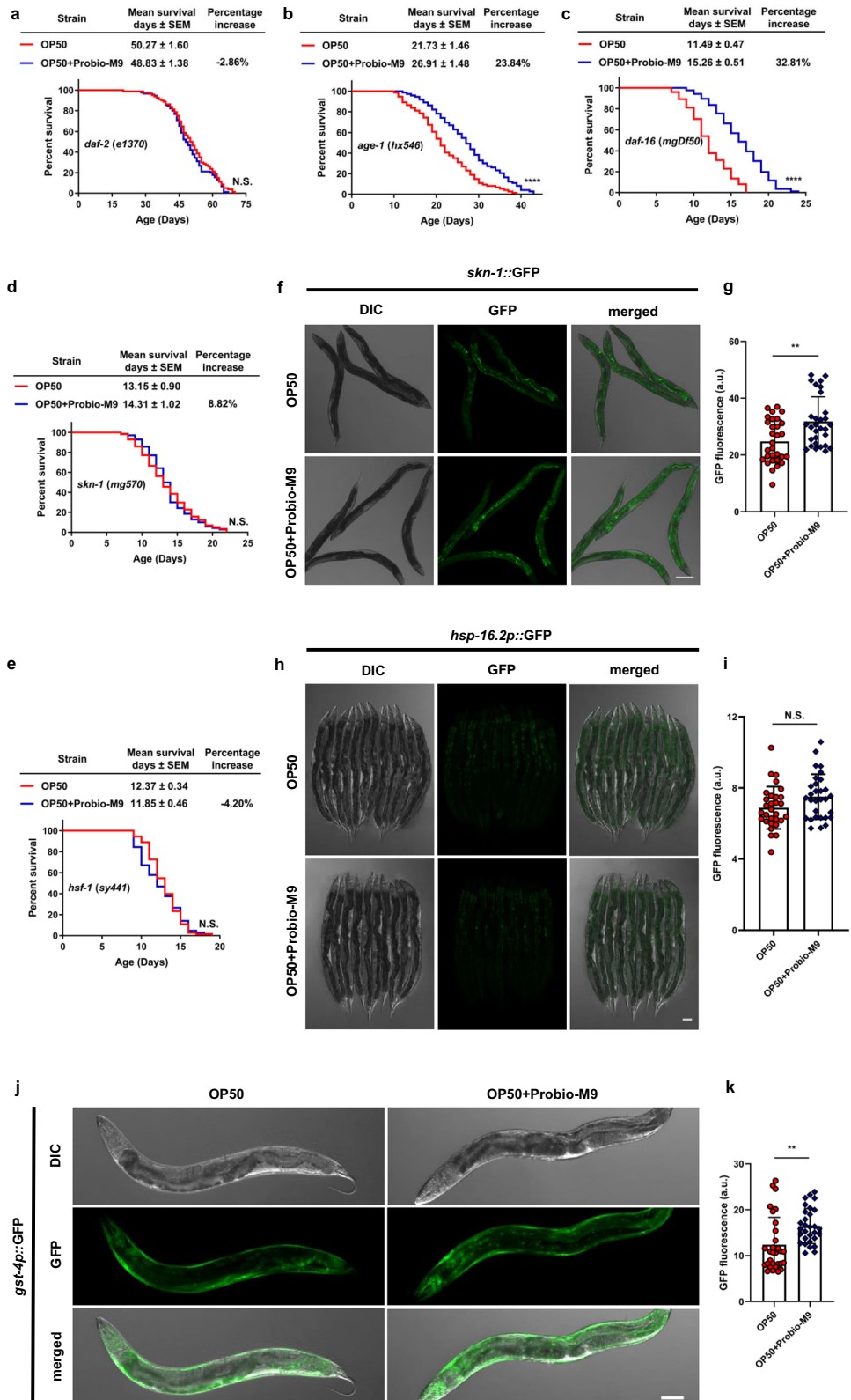

harmful bacteria. Several studies have shown that beneficial bacteria help to maintain the intestinal microbiome homeostasis and thereby promote host health[42,55,56]. However, when *E. coli* was used as a sole dietary supplementation in *C. elegans*, the increased numbers of *E. coli* in the worm intestine contributed to increased pathogenicity and accelerated aging in the worms[57]. Our study also showed that the extent of lifespan extension

conferred by Probio-M9 was dependent on the timing of Probio-M9 feeding. Furthermore, our quantitative CFU assay showed the continued presence of Probio-M9 in worms after switching from OP50 + Probio-M9 to OP50 feeding, similar to the positive control LGG. These results are also consistent with a previous report in which plate counting and transmission electron microscopy analysis were used to show that *Limosilactobacillus*

**Fig. 5 Probio-M9-induced lifespan extension is associated with the downstream effectors of insulin-like signaling pathway.** Lifespans of insulin-like signaling pathway mutants, *daf-2* (*e1370*) (**a**), *age-1* (*hx546*) (**b**), *daf-16* (*mgDf50*) (**c**), *skn-1* (*mg570*) (**d**) and *hsf-1* (*sy441*) (**e**) fed with Probio-M9 (*N* = 90 worms, *p* > 0.05, *p* < 0.0001, Log rank test). The stress response reporters *skn-1*::GFP (**f**) and *hsp-16.2p*::GFP (**h**) were used to measure SKN-1 and HSF-1, respectively. Day 5 worms were heated for 20 min at 34 °C, which induced *hsp-16.2p*::GFP reporter expression. **g**, **i** Fluorescence intensity of GFP was quantified using ImageJ software. The *skn-1*::GFP (**g**) worms grown on OP50 + Probio-M9 showed increased fluorescence intensity in the intestine (*N* = 30 worms, *p* < 0.01, Student's *t* test). Probio-M9 had no influence on HSF-1 (**i**) responding stress induction (*N* = 30 worms, *p* > 0.05, Student's *t* test). **j**, **k** The expression of GST-4, an indicator of SKN-1 activity, increased after OP50 + Probio-M9 feeding. Expression of *gst-4p*::*gfp* was detected on day 5 worms (**j**), and fluorescence intensity was measured using ImageJ software (**k**) (*N* = 30 worms, *p* < 0.01, Student's *t* test). In **f**, **h** and **j**, scale bar, 20 μm. In **g**, **i** and **k**, results are shown by arbitrary units (a.u.), values are presented as the mean ± SEM.

*fermentum* JDFM216 adhered to the intestinal tract of *C. elegans*[41]. However, OP50 + heat-inactivated Probio-M9 was unable to extend the lifespan of worms, suggesting that Probio-M9 in an active state has beneficial effects on host health that extend the lifespan of worms, consistent with the definition of probiotics[6]. By contrast, a previous report showed that B*ifido*bacterium *longeum* subsp. *longeum* BB68 that was heat-inactivated at 95°C for 30 min significantly extended the lifespan of worms by 28%[58], indicating that probiotics might have strain-specific properties. Altogether, our results indicate that Probio-M9 has beneficial effects on health-promoting and longevity in *C. elegans*, and these effects are dependent on adhesion of Probio-M9 to the worm intestine.

It is well known that aging is a complex process involving interactions among multiple signaling pathways[16,59,60], including the p38 signaling pathway, a key signaling pathway that regulates host lifespan[61]. Our results showed that OP50 + Probio-M9 was unable to extend the lifespan of loss-of-function *pmk-1* mutant, indicating that the p38 cascade might be involved in the longevity related to Probio-M9 feeding, consistent with a previous report of *Lacticaseibacillus rhamnosus* GG extending the lifespan of *C. elegans* by activating the *pmk-1* (p38 MAPK) signaling pathway[62]. The TIR-1 effector protein can activate the downstream PMK-1 signaling pathway[63,64]. Our data demonstrated that OP50 + Probio-M9 extended the lifespan of worms by activating TIR-1, upstream of the p38 signaling pathway. Related studies have also confirmed that *Lactobacillus acidophilus* NCFM activates the *C. elegans* immune response against *Enterococcus faecalis* and *Staphylococcus aureus* through TIR-1 and PMK-1 immune signaling pathways, and delays host aging as well[34]. Additionally, ATF-7, which acts downstream of the p38 signaling pathway, has a dual immunomodulatory effect on *C. elegans*. ATF-7 suppresses the transcription of immune-related genes. However, this inhibition is eliminated when p38 (PMK-1) phosphorylation occurs, after which ATF-7 acts as an activator, promotes the activation of immune genes[65]. We also observed that OP50 + Probio-M9 was unable to extend the lifespan of loss-of-function *atf-7* mutant, suggesting that the innate immune pattern recognition system might be involved in the Probio-M9-activated p38 cascade. Usually *E. coli* is used as the sole dietary supplementation for *C. elegans* culture; however, proliferating bacteria might be pathogenic to older worms[66]. Therefore, the ingested Probio-M9 might activate the host defense systems through PMK-1 to contribute its health-promoting effects. In a future investigation, we plan to compare the transcriptomes and metabolomes of *pmk-1* mutant and wild-type N2 worms fed with OP50 + Probio-M9, which may reveal genes and relevant metabolites regulated by *pmk-1* in contributing to retarding aging.

SKN-1 controls specific pathways of stress resistance and is also widely involved in immune response, metabolism, and detoxification[67]. Kato et al. showed that *Clostridium butyricum* MIYAIRI 588 could improve stress resistance and extend the lifespan of *C. elegans* via *skn-1*[68]. Our results illustrated that OP50 + Probio-M9 was unable to extend the lifespan of loss-of-function *skn-1* mutant, indicating that SKN-1 might be required

for the Probio-M9-mediated lifespan extension. SKN-1 has been shown to be directly suppressed by DAF-2[22], and DAF-2 signaling pathway via SKN-1 activity might regulate the lifespan of worms fed with Probio-M9. Bishop and Guarente revealed that the stress resistance function of SKN-1 was mediated by its expression in the intestine[48]. Our results showed that the intensity of fluorescence for SKN-1 in the intestine of worms was significantly stronger in OP50 + Probio-M9-fed compared with OP50. Besides, OP50 + Probio-M9 feeding increased the expression of *gst-4p*::*gfp* in *C. elegans*, the downstream target of *skn-1*[69], suggesting that Probio-M9 feeding activates SKN-1, which in turn contributes to improving stress resistance and extending the lifespan of *C. elegans*. Consistent with our findings, Komura et al. found that *Bifidobacterium infantis* feeding improves stress resistance and extends the lifespan of *C. elegans* via *skn-1*[70] signaling pathway.

Cellular organelles mitochondria were likely originated from bacteria; indeed, they share a number of metabolic pathways[71]. Mitochondria are involved in aging processes, as a signaling hub for both metabolism and the vital signaling pathways[72]. Our results demonstrated that Probio-M9 enhanced intestinal tract UPR^mt levels and extended lifespan of worms, probably through the mitochondrial ETC components ISP-1 and NUO-6. The reduced function of mitochondrial ETC complexes contributes to the activation of the UPR^mt to maintain mitochondrial homeostasis[73,74]. UPR^mt-activated ATFS-1 in turn inhibits ETC gene expression in the mitochondrial genome[75]. Consistent with our findings, Han et al. found that *E. coli* mutants with increased secretion of the metabolite colanic acid (CA) extended the lifespan of *C. elegans* by regulating host mitochondrial dynamics and the UPR^mt, and that purified CA polymers promoted the longevity of worms through ATFS-1[2]. Our results suggest that ATFS-1 is also a crucial mediator of the Probio-M9 health-promoting effect.

Bacterial metabolites play crucial roles in host aging[57,76,77]. Metabolomics analysis in this study found that the differentially expressed metabolites were mainly enriched in pathways involving amino acid metabolism, galactose metabolism, sphingolipid metabolism, fatty acid biosynthesis and cAMP signaling. Amino acids affect the longevity of the host, tryptophan and proline contribute to delaying the aging of *C. elegans*, while phenylalanine accelerates aging;[78] aspartate, glutamate and methionine extend the lifespan of yeast[79]. Our results suggested that the significant enrichment of bacterial tryptophan, aspartate and glutamate metabolites could be related with the extension of the lifespan of *C. elegans* by Probio-M9. Bacterial glucose metabolites, such as glucose, galactose and lactose, may also be associated with host longevity participate in anti-aging activities[57]. Our metabolomics data indicated that the significant enrichment of bacterial galactose metabolites may contribute to the extension of the lifespan of *C. elegans* by Probio-M9. Consistent with our findings, Zanni et al. found that galactose secreted by *Lactobacillus delbrueckii* subsp. *bulgaricus* might be associated with its beneficial effects on *C. elegans*[80]. The accumulation of bacterial metabolites such as sphingolipid and ceramide are closely related to aging and age-related diseases[81,82]. It has been

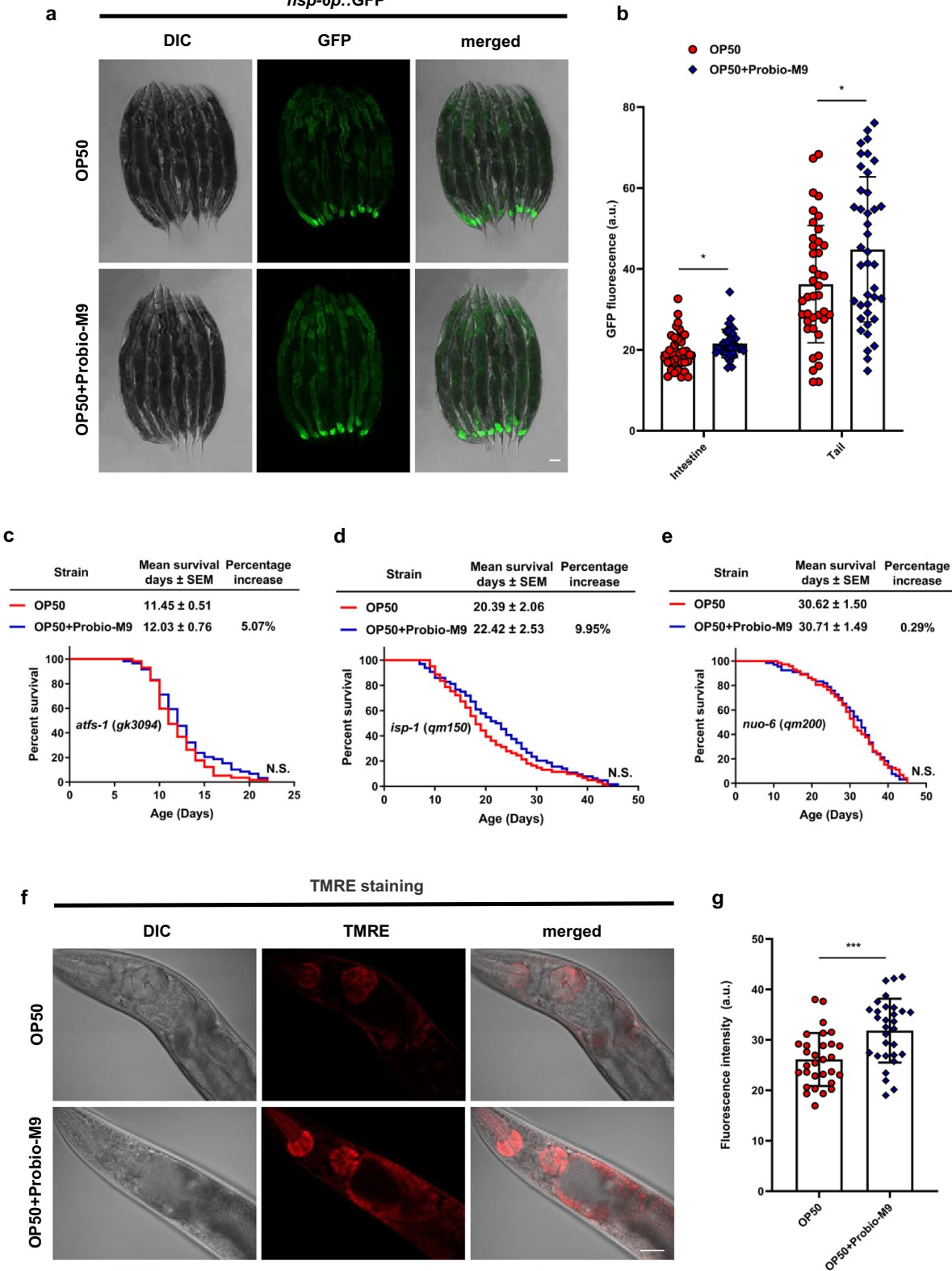

shown that sphingolipid metabolism might be vital for enhancing locomotor ability[83] and improvement of the health status of *C. elegans*, which were partially dependent on the insulin receptor *daf-2* signaling pathway[84]. Goya et al. observed that *Bacillus subtilis* mediated host sphingolipid metabolism and inhibited *a-syn* aggregation in *C. elegans*[85]. In light of our results, we revealed that Probio-M9 might regulate sphingolipid metabolism through

nutrient sensing to extend lifespan of *C. elegans*. Besides, our data showed that 2-aminoethanesulfonic acid derived from taurine metabolism was significantly accumulated upon OP50 + Probio-M9 feeding. Taurine is of great significance in the scavenging of lipid metabolism, immune function, and other physiological functions, making it an important promoter of anti-inflammatory activity[86,87]. However, such inference needs further experimental

**Fig. 6 Probio-M9 acts on host mitochondria to promote healthspan. a, b** UPR$^{mt}$ was detected by mitochondrial chaperone reporter *hsp-6p::gfp*. The representative photomicrographs of *hsp-6p*::GFP were shown in **a**, and the fluorescence intensity of *hsp-6p*::GFP was quantified using ImageJ software (**b**). The *hsp-6p*::GFP worms grown on OP50 + Probio-M9 showed increased ability to respond to stress in the intestine and tail ($N = 40$ worms, $p < 0.05$, Student's *t* test). **c** In the mutant of UPR$^{mt}$ transcription factor *atfs-1* (*gk3094*), the lifespan extension conferred by Probio-M9 is fully abolished ($N = 90$ worms, $p > 0.05$, Log rank test). **d, e** Probio-M9 is not able to extend the lifespan in the mutants of *C. elegans* ETC components, *isp-1* (*qm150*) (**d**) and *nuo-6* (*qm200*) (**e**) ($N = 90$ worms, $p > 0.05$, Log rank test). **f, g** The OP50 or OP50 + Probio-M9 fed worms were stained with TMRE to indicate the mitochondrial membrane potential. The pharyngeal bulbs were taken (**f**) and statistical analysis of TMRE fluorescence intensity in arbitrary units was shown in **g** ($N = 30$ worms, $p < 0.001$, Student's *t* test). In **a** and **f**, scale bar, 20 μm. In **b** and **g**, results are shown by arbitrary units (a.u.), values are presented as the mean ± SEM.

verification. It will be of interest to extract the most key metabolites in Probio-M9 and perform further lifespan verification experiments to demonstrate the health-promoting and longevity extension effects of probiotic metabolites on host. It is necessary to identify the major anti-aging components of probiotics and explore signaling pathways that probiotic utilizes to delay host aging. Although challenging, future study should focus on the effects of probiotics on longevity in mammals and elucidate the molecular mechanisms of action.

## Methods

***Caenorhabditis elegans* strains and culture**. The nematodes *C. elegans* and bacterial strains used in this study can be found in Supplementary Table 8. Worms were cultured at 20 °C on nematode growth medium (NGM) agar[88] plates seeded with pre-cultured bacterial strains according to standard techniques[89]. Hermaphrodites of *C. elegans* were age-synchronized by isolating eggs, and then transferred to NGM plates seeded with *E. coli* OP50[90].

**Culture conditions of bacterial strains**. In this study, Probio-M9 was isolated from healthy breast milk. *Lacticaseibacillus rhamnosus* GG (LGG) was acquired from Guangdong Microbial Culture Preservation Center. Probio-M9 and LGG were anaerobic cultured in MRS broth at 37 °C for 24 h and continuously cultured to three generations as test bacterial solution. *E. coli* OP50 was incubated in Luria-Bertani (LB) broth at 37 °C for 12 h with a shaking at 225 rpm.

For preparing bacterial plates for the worms feeding, Probio-M9 and LGG were collected through centrifugation at $4000 \times g$ for 10 min, and washed with M9 buffer (3 g $KH_2PO_4$, 6 g $Na_2HPO_4$, 5 g NaCl and 1 mL 1 M $MgSO_4$) twice, and then resuspended in *E. coli* OP50 ($6.8 \times 10^8$ CFU/mL) to a final dose ($2.82 \times 10^9$ CFU/mL and $3.02 \times 10^9$ CFU/mL, respectively), which were used as the bacteria supply. Before resuspending in *E. coli* OP50, the concentrated Probio-M9 was heated at 95 °C for 30 min and used as heat-inactived Probio-M9. 40 μL of the bacteria solution was inoculated on 60 mm NGM plate.

**Lifespan assay**. As described previously, all lifespan analyses were performed at 20 °C[88]. Late L4 stage worms were used as t = 0 for lifespan assay and transferred to fresh plates every two days until all worms died, unless otherwise noted. For each experiment, 90 worms were tested on 3 plates (30 worms per plate) for bacterial strain. Survival of the worms was determined on a daily basis. Ceasing pharyngeal pumping and not responding to gentle mechanical stimulation by platinum wire picker were scored death[2,53]. Worms, which accidentally lost, extruded organs or displayed matricidal hatching, were censored and excluded from lifespan analysis.

**Bacterial choice assay**. Worm eggs were cultured on OP50 or OP50 + Probio-M9 until they reached late L4 stage. And then these worms were washed with M9 buffer three times. For the first paradigm, OP50 or OP50 + Probio-M9 was seeded on the center of plate, and 50 worms were placed at the edges of each end of the plate, being equidistant from OP50 or OP50 + Probio-M9. For the second paradigm, OP50 or OP50 + Probio-M9 was seeded on the edges of each end of the plate, and 100 worms were placed in the center of the plate, being equidistant from OP50 and OP50 + Probio-M9. The worms were allowed to move freely for 1 h or 2 h, and the number of worms entering each bacterial lawn was counted.

**Life cycle, brood size and body size assays**. The assays were determined according to the previous method[55,91].

To determine the life cycle of worms, 20 eggs were placed on 20 plates and used as t = 0 for life cycle analysis. When the worm lays its first egg, which is the time to complete the life cycle.

To evaluate the brood size of worms, 20 synchronized hermaphrodites at late L4 stage were transferred to fresh plates daily (one worm per plate) until reproductive termination, the total number of offspring per worm was counted.

To measure the body size of worms, 20 worms were selected from day 1–5 of adulthood. The anaesthetized with 25 mM levamisole worms were taken photos

using a stereomicroscope (ZEISS Axio Imager M2), and analyzed using ImageJ software (National Institutes of Health, Bethesda, MD, USA). The projected area of worm was computed and used as an indicator of body size.

**Pharyngeal pumping rate assay**. The pharyngeal pumping rate of worms at different ages was examined according to the previous method[2]. Worms were selected from days 2–14 of adulthood and the pharyngeal pumping rate was measured using SMZ-168 stereo microscope (Motic). The times of pharyngeal constrictions in the terminal bulb of the pharynx for 30 s interval were counted.

**Locomotor scoring**. The motility of worms at different ages was measured according to the previous method[50]. Worms were selected from day 8–16 of adulthood and their locomotor scoring was measured. Classification of worms was based on difference in the movement activity. When worms show spontaneous and/or rhythmical sinusoidal movement (normal locomotion), they were classified as class "A"; worms with irregular and/or uncoordinated movement (uncoordinated/sluggish), they were classified as class "B"; worms moved merely their head and/or tail in respond to a moderate touch with a platinum wire picker (cannot move body), they were classified as class "C"; dead worms were classified as class "D". At least 90 worms were scored for each experiment.

**Lipofuscin accumulation assay**. The autofluorescence of intestinal lipofuscin was detected according to the previous method[52]. Day 10 or 15 worms were washed with M9 buffer three times, and then anaesthetized with 25 mM levamisole. Autofluorescence images of lipofuscin were obtained using blue excitation light (405 nm) in confocal laser scanning microscope (Olympus FV1200). The accumulation level of lipofuscin was detected by fluorescence intensity quantification using ImageJ software.

**Thermotolerance assay**. To assess thermotolerance, the survival rate of worms was measured according to the previous method[53]. Day 5 worms were exposed at 34 °C and then the survival rate was measured after 12 h, 18 h and 24 h, respectively.

**Determination of bacterial CFU within worm gut**. The number of CFU in the worm gut was determined according to the previous method[55]. Day 5 or 10 worms were washed three times with M9 buffer and transferred them on empty NGM plates for 1 h to remove surface-attached bacteria. 5 worms were placed into M9 buffer and ground with a glass grinder. The worm lysate was continuously diluted with M9 buffer, and anaerobic cultured at 37 °C for 48 h on MRS plates, the number of CFU was counted.

The number of bacteria pulse-chase in the worm intestine was determined according to the previous method[2]. Day 10 of adulthood worms were selected from the NGM plates lawned with OP50 + LGG (as a positive control) or OP50 + Probio-M9, washed with M9 buffer three times, and transferred them to the plate lawned with OP50. After 3 days, the number of CFU was tested as described above.

**Analysis of the fluorescence intensity of stress responses**. The fluorescence expression was measured according to the previous method[92]. Fluorescence intensity images were obtained using green excitation light (473 nm) of laser confocal scanning microscope (Olympus FV1200). Fluorescence intensity of GFP was analyzed using ImageJ software.

To measure *hsp-4p::gfp* fluorescence expression, day 5 worms were induced for 4 h in M9 buffer containing 50 ng/mL tunicamycin or tantamount DMSO. To measure *hsp-16.2p::gfp* fluorescence expression, day 5 worms were heated at 34 °C for 20 min, and then recovery for 6 h. At last, the fluorescence photos were taken and analyzed as described above.

**Estimation of mitochondrial membrane potential**. To evaluate the effect of mitochondrial membrane potential, a specific fluorescent dye tetra-methylrhodamine ethyl ester (TMRE) was used to stain worms. Worms were cultured in OP50 or OP50 + Probio-M9 containing 10 μm TMRE for 5 days, and

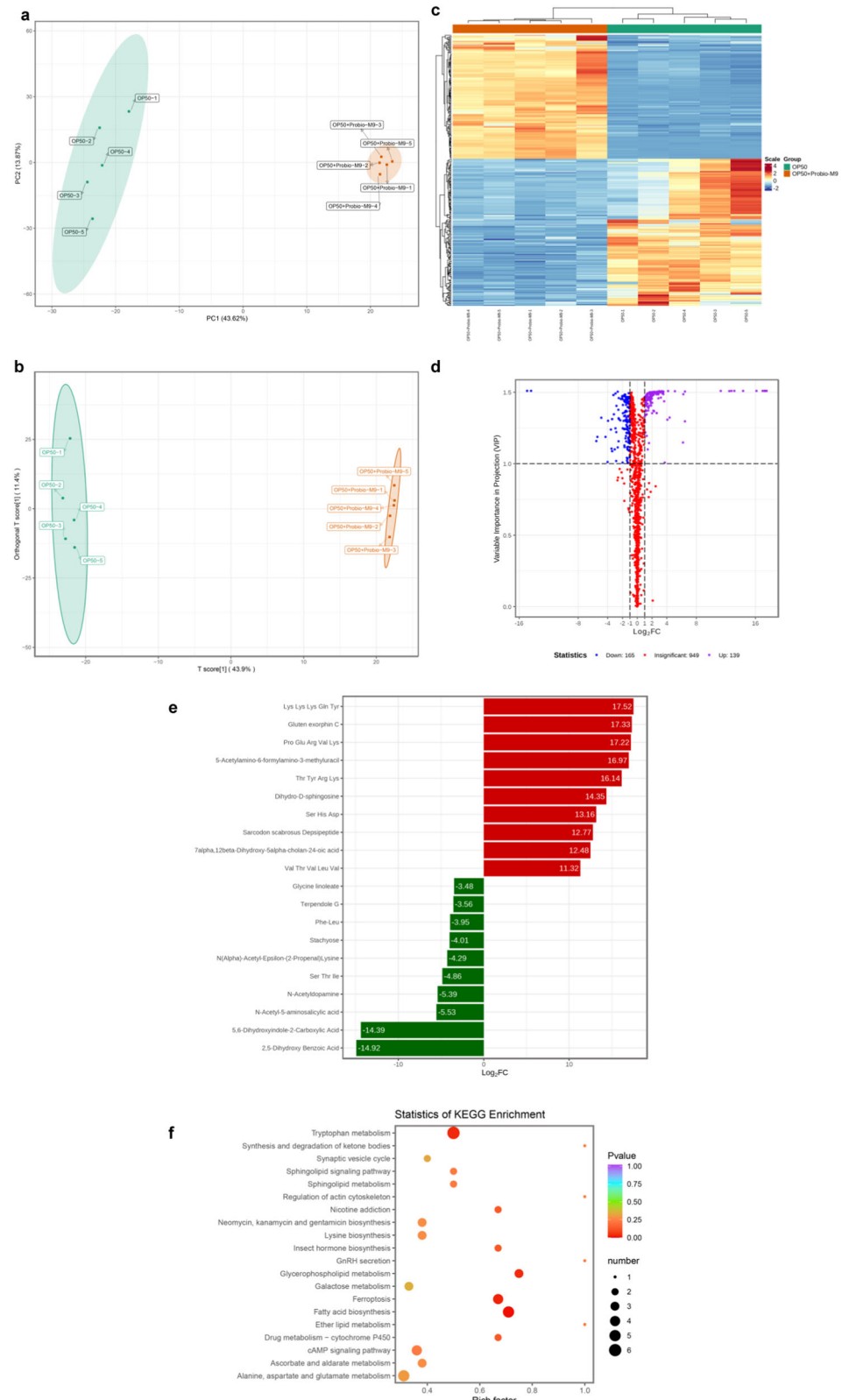

then placed on 2% agarose pads with 25 mM levamisole to take photos using red excitation light (556 nm) of laser confocal scanning microscope (Olympus FV1200). Fluorescence intensity of pictures was analyzed using ImageJ software.

**Metabolomics analysis**. The bacteria of control group (OP50) and experimental group (OP50 + Probio-M9) prepared for worms feeding were collected from NGM plates with M9 buffer to 15 mL centrifuge tube, washed with M9 buffer for three times, taken 50 ± 2 mg, added cold steel balls and homogenized at 30 Hz for 3 min.

The homogenized solution was added 1 mL 70% methanol with internal standard. Samples were vortexed for 2 min, incubated on ice for 15 min and then centrifuged (13,400 × g, 4 °C, 10 min). Samples were drawed 400 μL of supernatant into the EP tube and stored at −20 °C overnight. Samples were thawed on ice, centrifuged (13,400 × g, 4 °C, 3 min), and taken 200 μL of supernatant in the liner of the injection bottle for on-board analysis.

Metabolomics samples were analyzed using UPLC system (ExionLC AD, https://sciex.com.cn/) and Quadrupole-Time of Flight (TripleTOF 6600, AB SCIEX).

**Fig. 7 Comparison of the metabolic profiles between OP50 and Probio-M9.** PCA scores (**a**), OPLS-DA score plots (**b**) and heat map (**c**) of metabolites showed significant differences between OP50 and OP50 + Probio-M9, without overlap or crossover, with obvious separation. **d** Volcano plot displaying significantly metabolites between OP50 and OP50 + Probio-M9. Compared with OP50, purple dots represent metabolites that were increased with OP50 + Probio-M9 (up-regulated), blue dots represent metabolites that were decreased with OP50 + Probio-M9 (down-regulated), and red dots represent metabolites that were not changed with OP50 + Probio-M9 (insignificant). **e** The top ten changed metabolites between OP50 and OP50 + Probio-M9. Compared with OP50, red bars represent metabolites that were up-regulated by OP50 + Probio-M9, green bars represent metabolites that were down-regulated by OP50 + Probio-M9. **f** KEGG enrichment analysis of metabolites that were significantly changed between OP50 and OP50 + Probio-M9 (the color of the point represents *p* value, and the color is the redder, the enrichment degree is the more significant; the size of the point represents the number of enriched metabolites within this pathway).

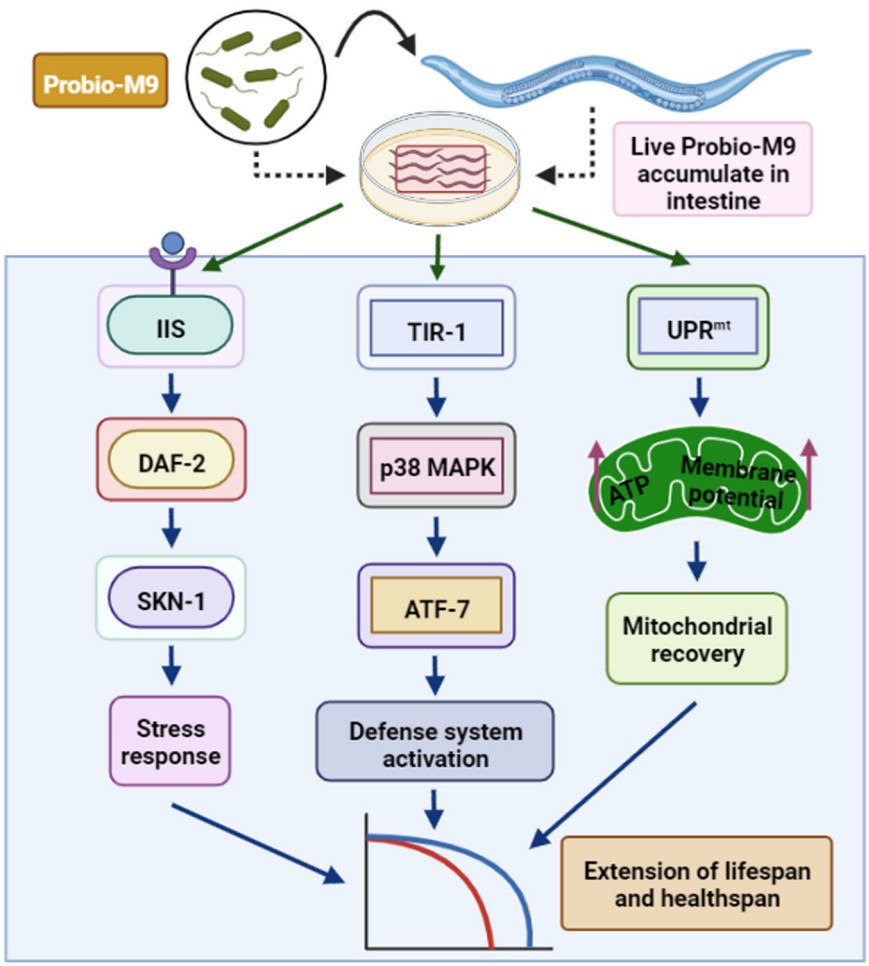

**Fig. 8 Schematic representation of signaling pathways regulated by Probio-M9 feeding in *C. elegans*.** Live Probio-M9 feeding increases the lifespan of worms via the Insulin/IGF-1 signaling (DAF-2/SKN-1) and p38 MAPK signaling (NSY-1–SEK-1–PMK-1) pathways. It also triggers mitochondrial unfolded protein response (UPR^mt) to maintain mitochondrial homeostasis. Therefore, Probio-M9 feeding extends the lifespan of *C. elegans* by improving the stress resistance, activating defensive system, and enhancing UPR^mt.

Samples were separated using a ACQUITY HSS T3 ($2.1 \times 100$ mm, 1.8 μm) at 40 °C. Mobile phases were consisting of mobile phase A (water with 0.1% formic acid, v/v) and mobile phase B (acetonitrile with 0.1% acetic acid, v/v). The elution gradient was as follows: 95% A to 5% B for 0.0 min; 10% A to 90% B for 10.0 min; 95% A to 5% B for 14.0 min. The flow rate was 0.35 mL/min with an injection volume of 5 μL.

Software Analyst 1.6.3 was used to process the mass spectrum data. The repeatability of metabolite extraction and detection were estimated using TIC and multi peak multiple reaction monitoring (MRM). According to home-made metadata database (MWDB) and other databases, qualitative analysis of information and secondary general data was carried out based on retention time (RT) and mass-to-charge ratio. The metabolite structure analysis referred to several public mass spectrometry databases, including Human Metabolome Database (HMDB, http://www.hmdb.ca/), Metlin (http://metlin.scripps.edu/index.php), massbank (http://www.massbank.jp/) and knapsack (http://kanaya.naist.jp/knapsack/).

Ten microliters per sample was used for quality control (QC) of metabolomics analysis. When running sample sets on column, one QC sample was injected after 10 samples. Metabolite quantification was analyzed using MRM of triple quadrupole mass spectrometry. The mass spectrum file under the sample machine was opened with multi-quantitative software, and the chromatographic peaks were integrated and calibrated. The peak area of each chromatographic peak represented the relative content of the corresponding substance. The integral data of chromatographic peak area was extracted and saved, and the positive and negative ions of metabolites were removed using self-built software package.

The coefficient of variation (CV) values of metabolites in QC samples were calculated, and metabolites with CV values greater than 0.5 were removed. When metabolites were detected in both positive and negative ionization modes, those with larger CVs were removed in either positive or negative mode. Orthogonal partial least squares discriminant analysis (OPLS-DA) model was used to identify differences in metabolic profiles between groups.

**Statistics and reproducibility**. All results were derived from at least three biologically replicates. Prism8 software was used to analyze the lifespan curves, and Log rank (Mantel–Cox) method was used to analyze the significance of the difference. For

statistical data, Student's *t* test, Chi-squared test, one-way ANOVA and two-way ANOVA analysis coupled with multiple comparisons were used for significance analysis. In all cases, $p < 0.05$ was considered significant. In the figures, the asterisk indicates the statistical significance of the Log rank test, Student's *t* test, Chi-squared test, one-way ANOVA and two-way ANOVA analysis as compared to control. For metabolomics data, the OPLS-DA model was applied using R package "MetaboAnalyst". Permutation test repeated 200 times were used to verify this model, and $p < 0.05$ indicated the available OPLS-DA model. The significance of each metabolite was measured by Student's *t* test and fold change, and one-tailed Student's *t* test or Fisher's exact test were used to analyze the statistical significance, $p < 0.05$ was considered to be statistically significant. Using the Benjamini–Hochberg method, *p* value was corrected for multiple testing by false-discovery rate (FDR).

**Reporting summary**. Further information on research design is available in the Nature Research Reporting Summary linked to this article.

## Data availability

The source data used to create the box plots in the figures were deposited in Supplementary Data 2. The source data used to create the survival curves in the figures were deposited in Supplementary Data 3. The source data used to create the differentially metabolites between OP50 and Probio-M9 in the figures were deposited in Supplementary Data 4.

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

## Acknowledgements

We are greatly thankful to professor Long Miao for his helpful comments and manuscript editing. We are grateful to professors Ye Tian and Xiaoyun Xu for their assistance with the worm strains. This work was financially supported by the Inner Mongolia Science and Technology Major Projects (2021ZD0014) and China Agriculture Research System of MOF and MARA and Natural Science Foundation of China (32070694, 31872822 to Y.Z.). Some worm strains were obtained from the Caenorhabditis Genetics Center (CGC, https://cgc.umn.edu/), which is funded by the NIH Office of Research Infrastructure Programs (P40 OD010440), USA.

## Author contributions

Experiments were designed by J.Z., Y.Z., Z.S., and T.S. All experiments were performed by J.Z. J.Z. wrote the manuscript with significant contributions from Y.Z., Z.S., and T.S. All authors made comments and suggestions on the manuscript.

## Competing interests

The authors declare no competing interests.
