## [Peer Review File · Communications Biology]

Reviewers' comments:

Reviewer #1 (Remarks to the Author):

In this study, Zhang et al. have performed a comprehensive characterization of the pro-longevity effects of the probiotic bacteria *Lactocaseibacillus rhamnosus* Probio-M9 on the lifespan of the nematode *Caenorhabditis elegans*. They have performed detailed genetic analyses to identify the genes that might contribute to the lifespan extending effects of Probio-M9. In addition, they have also qualified several metabolic and physiological parameters of Probio-M9-fed animals. The magnitude of lifespan extension produced via Probio-M9 feeding (~30%) will be of great interest to the community of aging researchers because this effect is comparable to that observed for the most robust pharmacological interventions. Secondly, the fact that the pro-longevity effects of this bacteria requires attachment of live bacteria to the intestine is particularly interesting for the design of probiotic treatments, since heat-killed Probio-M9 produces no beneficial effects.

The data in this manuscript is mostly of very good quality, but the statistical testing and its reporting are not adequate for multiple experiments. In addition, the proposed role of *daf-2* signaling via *skn-1* and *hsf-1* to mediate the longevity promoting effects of Probio-M9 needs to be supported with more experimental evidence.

Major comments:

1. One very interesting but also confusing finding in this study is that the lifespan extension effect of Probio-M9 is dependent on the insulin receptor *daf-2*, but not on its downstream targets *age-1* and *daf-16*. In the context of dauer formation and lifespan regulation, *daf-2*, *age-1* and *daf-16* have been shown to act in the same pathway (PMIDs: 8601482, 9504918). Signaling via *daf-2* (insulin/IGF-1 receptor) activates *age-1* (PI3K), which via a series of phosphorylation events regulates the activity of *daf-16* (FoxO transcription factor). In this study, the authors observed that Probio-M9 can extend the lifespan of *age-1* and *daf-16* mutants, but not the *daf-2(e1370)* mutant. They propose that *daf-2* is involved in extending the lifespan of Probio-M9 fed animals via activating the stress-associated transcription factors *skn-1* and *hsf-1*. Previous studies have shown that *skn-1* is required for the lifespan extension phenotypes of *daf-2(e1370)* mutant worms only at 15 C but not at 20 C (PMID: 25517099). However, since the authors performed all their lifespan studies at 20 C, it is unlikely that the *daf-2*-dependent pro-longevity effects of Probio-M9 were mediated via *skn-1*. Hence, to convincingly show that Probio-M9 activates the *skn-1* and *hsf-1* pathways downstream of *daf-2* signaling, but not the *age-1* and *daf-16*-dependent pathways, the authors need to provide some evidence of activation of *skn-1* and *hsf-1* after feeding Probio-M9. Surprisingly, the authors did not observe any significant increase in the expression of *hsp-16.2* (Supplementary Fig. 3c,3d), a direct transcriptional target of *hsf-1*. The authors should use either qRT-PCR or existing *gfp* reporters for known transcriptional targets of *skn-1* and *hsf-1* to test whether the expression of any of those genes is upregulated after Probio-M9 feeding.

2. In the abstract, the authors mention that Probio-M9 extends lifespan via activating the *daf-2* signaling pathway. However, the first identified and widely accepted role of *daf-2* signaling in extending lifespan is via *age-1* and *daf-16* (PMID: 9504918), but the authors show that neither *age-1* nor *daf-16* are required for the lifespan extension associated with Probio-M9. Hence, to avoid confusion among the readers, the authors should clarify in the abstract that Probio-M9 extends lifespan via *daf-2* signaling that is independent of *daf-16*, but dependent on *skn-1* and *hsf-1*.

3. Fig. 2e: The authors show that Probio-M9 fed worms show higher pumping rates at older ages. However, it is not possible to conclude whether Probio-M9 'delays' the age-associated rate of decline in pharyngeal pumping in worms unless the pumping rate of worms in early adulthood is provided. For example, if the pumping rate of worms on Probio-M9 is higher than that of OP50 control even in young adult worms, this would mean that pharyngeal pumping rate declines at the same rate in OP50-fed and Probio-M9-fed worms. To interpret these data, the pharyngeal pumping data for these two conditions should also be measured in young adult worms (day 1, day 2, day 4). In addition, multiple pairwise t-tests without p-value correction for multiple comparisons results in false-positive findings and is also not appropriate for this analysis. The authors need to perform a two-way ANOVA to show that in addition to the effects of 'diet' and 'age', the interaction term 'diet X age' also has a significant effect on pharyngeal pumping. Only then it can be concluded that Probio-M9 delays age-associated decline in pharyngeal pumping in worms.

4. Page 10: "Interestingly, we found that OP50+Probio-M9 was completely unable to extend the lifespans of *nsy-1(ag3)*, *sek-1(km4)* or *pmk-1(km25)* mutant worms" – the authors should interpret these data carefully. For the *sek-1(km4)* mutant, Probio-M9 consistently produces 10% or higher lifespan extension across the three independent replicates (9.9%, 11.6% and 13.3%). Though the effects were not found to be statistically significant, the effect is consistent across repeats, which suggests biological significance. Hence, the authors should describe that while *nsy-1* and *pmk-1* mutants show almost complete suppression of the lifespan extension phenotype associated with Probio-M9 feeding, *sek-1* shows strong, but not complete suppression of the longevity phenotype (~30% lifespan extension in N2 vs ~12% extension in the *sek-1* mutant).

5. Page 12: "OP50+Probio-M9 was unable to extend the lifespan of the *isp-1(qm150)* or *nuo-6(qm200)* mutants, suggesting that the ETC in mitochondria is indispensable for the lifespan-extending property of Probio-M9." The authors need to interpret these data carefully. Firstly, both *isp-1(qm150)* or *nuo-6(qm200)* are not null mutants, but instead are hypomorphs due to substitution of a single amino acid in these proteins. Hence, it cannot be concluded that ETC proteins are indispensable for the lifespan extension phenotype of Probio-M9. Secondly, both *isp-1(qm150)* or *nuo-6(qm200)* are extremely long-lived compared to wild-type N2 animals. Hence, the most likely interpretation of the data reported in Figs 6d,e is that the lifespan extending effects of Probio-M9 converge with the pro-longevity effects of reduced mitochondrial function in worms. The authors should also change the claim that "Probio-M9 extends lifespan likely through interaction with the mitochondrial ETC complexes".

Minor comments:

- a) Abstract: "Probio-M9 protects and repairs damaged the mitochondria" – replace with 'Probio-M9 protects and repairs damaged mitochondria'.
- b) The authors should cite the study (PMID: 35127565) that has previously suggested a role of the *pmk-1* (p38 MAPK) signaling pathway in extending lifespan of *C. elegans* fed with the GG strain of *Lactocaseibacillus rhamnosus*.
- c) Page 4: "*daf-2* is a key upstream component of IIS, regulating various physiological processes, such as aging and adult lifespan in *C. elegans*" – Here the authors should also cite the first study that showed the longevity phenotype of *daf-2* mutants (PMID: 8247153). In addition, they should describe that the lifespan extending effects of *daf-2* are mediated via the *daf-16/FoXO* transcription factor in reduced insulin signaling mutants such as *daf-2(e1370)*.
- d) Page 4: "*SKN-1*, a stress-responsive nuclear respiratory factor" – replace 'respiratory factor' with 'transcription factor'.
- e) Page 7: The authors claim that Probio-M9 extends lifespan in a dietary restriction (DR)-independent manner. They make this claim based on findings that feeding worms with Probio-M9 does not produce any significant effect on developmental rate, brood size or body length. To conclusively demonstrate that Probio-M9 does not extend lifespan via dietary restriction, the authors should also discuss the results for the very well-studied genetic model of DR, the *eat-2(ad1116)* strain, in this section (Supplementary Fig. 2c).
- f) In Supplementary Figs 1b and 1d, the y-axes represent percentage. Hence, error bars above 100% are misleading. In addition, t-tests are not appropriate for comparison of proportions. The authors should use Chi-squared test for proportions instead.
- g) In the Supplementary Tables for lifespan data, the authors should also list the total number of animals for each condition/genotype and how many of those were censored during the lifespan analysis.
- h) In the Supplementary Tables for lifespan data, the actual p-value should be listed instead of 'p > 0.05'. This is because in Supplementary Table 3, lifespan extension up to 13% has been reported as non-significant, while in Supplementary Table 2, a 7% lifespan extension is reported as statistically significant. The actual p-value for these conditions needs to be reported to know whether the effects reported as 'non-significant' were approaching significance (p < 0.1).
- i) In Fig. 1b, 'OP50+Probio-M9' should be written as 'OP50+4 Probio-M9'. Otherwise, the reader might assume that these two bacteria were present in a 1:1 ratio (based on the other conditions used in this experiment).
- j) Fig. 2f: What statistical tests were used for these data? Chi-squared tests for proportions should be used and the p-values need to be corrected for multiple comparisons.
- k) Figs 2h, 2i: Were the p-values for the t-tests corrected for multiple comparisons?
- l) Page 9: The authors make the claim that "We also found that worms cultured from eggs to adulthood (day 2 or day 4 after L4 stage) on OP50+Probio-M9 had a significantly longer lifespan

extension that those cultured from eggs to the L4 stage on OP50+Probio-M9". However, they did not perform the statistical analysis for this comparison. Since they performed all these lifespan experiments in triplicates, they can perform statistical tests on the lifespan extension values for the three types of treatments. One-way ANOVA followed by post-hoc tests would be ideal for this comparison.

m) Fig. 7c – The color of this heatmap should be changed to red-blue format to make it comprehensible for colorblind individuals.

Reviewer #2 (Remarks to the Author):

In this manuscript, Zhang et al investigate the effect of *L. rhamnosus* Probio-M9 supplementation on lifespan extension in a model nematode, *C. elegans*. They also elucidated molecular mechanism and metabolic changes underlying the effect of probiotic bacteria on the lifespan extension. So far, the probiotic bacteria-mediated lifespan extension is already known in the previous research, but authors elucidate detailed molecular pathway of the probiotics effects such as p38 and daf-2 signaling, mitochondrial unfolded protein response. One of main novel point of the present investigation is the metabolite analysis. Research topic of present study seems proper for Communications Biology. Authors careful design and execute experiment to support their hypothesis. They provide experimental data using various mutant *C. elegans* strain, and provide some novel knowledge to researchers who develop nutraceuticals for antiaging purpose. However, authors need to consider and answer the following points.

- 1) Authors analyze metabolic changes of *C. elegans* after the supplementation of probiotic bacteria by using UPLC analysis. But the detailed methods how authors identify and quantify each metabolite, therefore, authors need to describe the metabolomics analysis method more in detail in the Methods section.
- 2) Author just list up differently present metabolite and analyze the KEGG pathway. However, these seem not enough for the present study. Author need to execute some supplementation experiment of the most important metabolites and prove the functionality or causality of metabolite related to the lifespan modulation effects of probiotics. Author need to discuss meaning of important metabolites related to aging and anti-aging, based on the previous research.
- 3) In Abstract, author need to precisely describe change of metabolism is originated from *C. elegans* or probiotic bacteria itself. Last sentence of Abstract should be revised.
- 4) First paragraph of Result section, 1:4 ratio of bacterial mixture=> 2.82 or 2.72×10^9 CFU/ml of Probio-M9?

2 August 2022

Zhang et al. *Communications Biology* Tracking #:

COMMSBIO-22-0589-T

Response to reviewer comments

We are pleased to hear from you. We appreciate your interest, time and labor on our manuscript. We are most grateful for the positive and constructive comments provided by two reviewers and we revised manuscript accordingly in order to address their concerns.

We are pleased that both reviewers appreciate the significance of our study. To address the concern (the role of *daf-2* signaling via *skn-1* and *hsf-1* to mediate the longevity promoting effects of Probio-M9) raised by Reviewer #1, we added new data (Fig. 5f, g, j, k), adjusted the previous Supplementary Fig. 3c, 3d into the current Fig. 5h, i, and added the discussion of the role of *daf-2* signaling via *skn-1* to mediate the longevity promoting effects. Please see line 246-257 and line 425-441 of the Article.

We also added the discussion of important metabolites related to aging and anti-aging based on the previous research, as suggested by Reviewer #2, please see line 458-487 of the Article. Considering these suggested changes, we revised or rewrote the Abstract, Introduction, Results, Discussion, Methods, Figures and Figure legends sections. Our point-by-point responses to the reviewer comments are provided below.

The reviewers' comments are provided in black Roman typeface and our responses are written in red Roman typeface after each comment.

We are grateful to your comments and suggestions. We believe that the revised manuscript has been strengthened considerably and hope that it will now be considered suitable for publication in *Communications Biology*.

Sincerely,

Tiansong Sun

College of Food Science and Engineering

Key Laboratory of Dairy Biotechnology and Engineering, Ministry of Education

Inner Mongolia Agricultural University

Hohhot 010018

Inner Mongolia, China

Email: sts9940@sina.com

Reviewer #1 (Comments to the Author):

Thank you for your kind remarks and positive advice.

In this study, Zhang et al. have performed a comprehensive characterization of the pro-longevity effects of the probiotic bacteria *Lacticaseibacillus rhamnosus* Probio-M9 on the lifespan of the nematode *Caenorhabditis elegans*. They have performed detailed genetic analyses to identify the genes that might contribute to the lifespan extending effects of Probio-M9. In addition, they have also qualified several metabolic and physiological parameters of Probio-M9-fed animals.

The magnitude of lifespan extension produced via Probio-M9 feeding (~30%) will be of great interest to the community of aging researchers because this effect is comparable to that observed for the most robust pharmacological interventions.

Secondly, the fact that the pro-longevity of effects of this bacterium requires attachment of live bacteria to the intestine is particularly interesting for the design of probiotic treatments, since heat-killed Probio-M9 produces no beneficial effects.

The data in this manuscript is mostly of very good quality, but the statistical testing and its reporting are not adequate for multiple experiments. In addition, the proposed role of *daf-2* signaling via *skn-1* and *hsf-1* to mediate the longevity promoting effects of Probio-M9 needs to be supported with more experimental evidence.

Thank you for your kind remarks and helpful advice, which have greatly improved the quality of our manuscript. We will respond to your comments one by one. Please check the specific reply below.

Major comments:

1. One very interesting but also confusing finding in this study is that the lifespan extension effect of Probio-M9 is dependent on the insulin receptor *daf-2*, but not on its downstream targets *age-1* and *daf-16*. In the context of dauer formation and lifespan regulation, *daf-2*, *age-1* and *daf-16* have been shown to act in the same pathway (PMIDs: 8601482, 9504918). Signaling via *daf-2* (insulin/IGF-1 receptor) activates *age-1* (PI3K), which via a series of phosphorylation events regulates the activity of *daf-16* (FoxO transcription factor). In this study, the authors observed that Probio-M9 can extend the lifespan of *age-1* and *daf-16* mutants, but not the *daf-2* (*e1370*) mutant. They propose that *daf-2* is involved in extending the lifespan of Probio-M9 fed animals via activating the stress-associated transcription factors *skn-1* and *hsf-1*. Previous studies have shown that *skn-1* is required for the lifespan extension phenotypes of *daf-2* (*e1370*) mutant worms only at 15 °C but not at 20 °C (PMID: 25517099). However, since the authors performed all their lifespan studies at 20 °C, it is unlikely that the *daf-2*-dependent pro-longevity effects of Probio-M9 were mediated via *skn-1*. Hence, to convincingly show that Probio-M9 activates the *skn-1* and *hsf-1* pathways downstream of *daf-2* signaling, but not the *age-1* and *daf-16*-dependent pathways, the authors need to provide some evidence of activation of *skn-1* and *hsf-1* after feeding Probio-M9. Surprisingly, the authors did not observe any significant increase in the expression of *hsp-16.2* (Supplementary Fig. 3c,3d), a direct transcriptional target of *hsf-1*. The authors should use either qRT-PCR or existing gfp reporters for known transcriptional targets of *skn-1* and *hsf-1* to test

whether the expression of any of those genes is upregulated after Probio-M9 feeding.

Response: Thank you for your kind comment and suggestion. We understand the Reviewer's confusion that the lifespan extension effects of Probio-M9 is dependent on the insulin receptor *daf-2*, but independent on the general *daf-16*. Probio-M9 failed to extend the lifespan of *daf-2* (*e1370*) mutant, but significantly extended the *daf-16* (*mgDf50*) mutant similarly to that observed in wild-type *C. elegans*. This result appears contradictory to the extension effect of Probio-M9 observed for the *daf-16* mutant; however, this discrepancy is likely because the IIS pathway (DAF-2 is the only insulin/IGF-1 receptor) regulates SKN-1, which modulates stress responses in *C. elegans* (Reference 22). It is consistent with a previously report that *Bifidobacterium infantis* extends the lifespan of *C. elegans* by activating *daf-2* via *skn-1*, but not *daf-16* signaling pathway (Reference 68).

DAF-2 is involved in extending the lifespan of Probio-M9 fed animals via activating the stress-associated transcription factors *skn-1* and *hsf-1*, we have endeavored to provide evidence for this issue. We added new data of the expression of *skn-1::gfp* (transcriptional target of *skn-1*) and *gst-4p::gfp* (downstream target of *skn-1*) after feeding Probio-M9 and adjusted the previous figure of the expression of *hsp-16.2p::gfp* (a direct transcriptional target of *hsf-1*). Please see line 246-257 and Fig. 5f-k of the Article. Both the figures and figure legends are now updated.

To confirm whether Probio-M9 feeding improves stress resistance and extends the lifespan of *C. elegans*, we investigated two GFP reporters *skn-1::gfp* (transcriptional target of *skn-1*) and *hsp-16.2p::gfp* (a direct transcriptional target of *hsf-1*) to reflect

the stress resistance SKN-1 and HSF-1, respectively. Our results showed that the expression of *skn-1::gfp* increased in the intestine of worms fed with OP50+Probio-M9, but had no effect on *hsp-16.2p::gfp* (Fig. 5f–i). Furthermore, we detected the expression of GST-4, an indicator of SKN-1 activity (Reference 37), and the results indicated that the expression of *gst-4p::gfp* increased in *C. elegans* fed with OP50+Probio-M9, compared with those fed with OP50 (Fig. 5j, k). Taken together, these results suggested that Probio-M9 influences lifespan extension and stress resistance by a *skn-1*-dependent but *hsf-1*-independent mechanism.

We also added the discussion of the role of *daf-2* signaling via *skn-1* to mediate the longevity promoting effects. Please see line 425-441 of the Article.

SKN-1 controls specific pathways of stress resistance and is also widely involved in immune response, metabolism, and detoxification (Reference 67). Kato et al. showed that *Clostridium butyricum* MIYAIRI 588 could improve stress resistance and extend the lifespan of *C. elegans* via *skn-1* (Reference 68). Our results illustrated that OP50+Probio-M9 was unable to extend the lifespan of loss-of-function *skn-1* mutant, indicating that SKN-1 might be required for the Probio-M9-mediated lifespan extension. SKN-1 has been shown to be directly suppressed by DAF-2 (Reference 22), and DAF-2 signaling pathway via SKN-1 activity might regulate the lifespan of worms fed with Probio-M9. Bishop and Guarente revealed that the stress resistance function of SKN-1 was mediated by its expression in the intestine (Reference 48). Our results showed that the intensity of fluorescence for SKN-1 in the intestine of worms was significantly stronger in OP50+Probio-M9-fed compared with OP50. Besides,

OP50+Probio-M9 feeding increased the expression of *gst-4p::gfp* in *C. elegans*, the downstream target of *skn-1* (Reference 69), suggesting that Probio-M9 feeding activates SKN-1, which in turn contributes to improving stress resistance and extending the lifespan of *C. elegans*. Consistent with our findings, Komura et al. found that *Bifidobacterium infantis* feeding improves stress resistance and extends the lifespan of *C. elegans* via *skn-1* (Reference 70) signaling pathway.

- 22 Tullet, J. *et al.* Direct inhibition of the longevity-promoting factor SKN-1 by insulin-like signaling in *C. elegans*. *Cell* **132**, 1025-1038 (2008).
- 37 Leiers *et al.* A stress-responsive glutathione S-transferase confers resistance to oxidative stress in *Caenorhabditis elegans*. *Free Radical Biology and Medicine* **34**, 1405-1415 (2003).
- 48 Bishop, N. A. & Guarente, L. Two neurons mediate diet-restriction-induced longevity in *C. elegans*. *Nature* **447**, 545-549 (2007).
- 67 Blackwell, T. K., Steinbaugh, M. J., Hourihan, J. M., Ewald, C. Y. & Isik, M. SKN-1/Nrf, stress responses, and aging in *Caenorhabditis elegans*. *Free Radical Biology and Medicine*, 290-301 (2015).
- 68 Komura, T., Ikeda, T., Yasui, C., Saeki, S. & Nishikawa, Y. Mechanism underlying prolongevity induced by *bifidobacteria* in *Caenorhabditis elegans*. *Biogerontology* **14**, 73-87 (2013).
- 69 Ransome, V. D. H., Mccallum, K. C., Cruz, M. R., Garsin, D. A. & Ausubel, F. M. Ce-Duox1/BLI-3 generated reactive oxygen species trigger protective SKN-1 activity via p38 MAPK signaling during infection in *C. elegans*. *PLoS Pathogens* **7**, e1002453 (2011).
- 70 Komura, T., Ikeda, T., Yasui, C., Saeki, S. & Nishikawa, Y. Mechanism underlying prolongevity induced by *bifidobacteria* in *Caenorhabditis elegans*. *Biogerontology* **14**, 73-87 (2013).

2. In the abstract, the authors mention that Probio-M9 extends lifespan via activating the *daf-2* signaling pathway. However, the first identified and widely accepted role of *daf-2* signaling in extending lifespan is via *age-1* and *daf-16* (PMID: 9504918), but the authors show that neither *age-1* nor *daf-16* are required for the lifespan extension associated with Probio-M9. Hence, to avoid confusion among the readers, the authors should clarify in the abstract that Probio-M9 extends lifespan via *daf-2* signaling that is independent of *daf-16*, but dependent on *skn-1* and *hsf-1*.

Response: Thank you for your kind comment. We do understand and agree with the concern of this Reviewer, and we have corrected it. Please see line 35-37 of the Article.

By screening various aging-related mutants of *C. elegans*, we find that Probio-M9 extends lifespan via p38 cascade and *daf-2* signaling pathways, independent on *daf-16* but dependent on *skn-1*.

3. Fig. 2e: The authors show that Probio-M9 fed worms show higher pumping rates at older ages. However, it is not possible to conclude whether Probio-M9 ‘delays’ the age-associated rate of decline in pharyngeal pumping in worms unless the pumping rate of worms in early adulthood is provided. For example, if the pumping rate of worms on Probio-M9 is higher than that of OP50 control even in young adult worms, this would mean that pharyngeal pumping rate declines at the same rate in OP50-fed and Probio-M9-fed worms. To interpret these data, the pharyngeal pumping data for these two conditions should also be measured in young adult worms (day 1, day 2,

day 4). In addition, multiple pairwise t-tests without p-value correction for multiple comparisons results in false-positive findings and is also not appropriate for this analysis. The authors need to perform a two-way ANOVA to show that in addition to the effects of 'diet' and 'age', the interaction term 'diet X age' also has a significant effect on pharyngeal pumping. Only then it can be concluded that Probio-M9 delays age-associated decline in pharyngeal pumping in worms.

Response: Thank you for your kind comment and suggestion. We agree with the Reviewer's thoughts on the data of Probio-M9 delaying the decrease in pharyngeal pumping rate at older ages (days 6-14 of adulthood). To carefully interpret these results, we added the pharyngeal pumping rate of young adulthood worms (day 2 and day 4) in both conditions and performed a two-way ANOVA to show that in addition to the effects of 'diet' and 'age', the interaction term 'diet X age' also has a significant effect on pharyngeal pumping. Please see line 143-147 and Fig. 2g of the Article.

Both the figures and figure legends are now updated.

The pharyngeal pumping rate declined progressively with aging in both groups of worms, and feeding with OP50+Probio-M9 had no significant difference in pharyngeal pumping rate on days 2 and 4 of young adulthood, but significantly delayed the decrease in pharyngeal pumping rate on days 6-14 of adulthood (Fig. 2g).

4. Page 10: “Interestingly, we found that OP50+Probio-M9 was completely unable to extend the lifespans of *nsy-1* (*ag3*), *sek-1* (*km4*) or *pmk-1* (*km25*) mutant worms” – the authors should interpret these data carefully. For the *sek-1* (*km4*) mutant, Probio-M9 consistently produces 10% or higher lifespan extension across the three independent replicates (9.9%, 11.6% and 13.3%). Though the effects were not found to be statistically significant, the effect is consistent across repeats, which suggests biological significance. Hence, the authors should describe that while *nsy-1* and *pmk-1* mutants show almost complete suppression of the lifespan extension phenotype associated with Probio-M9 feeding, *sek-1* shows strong, but not complete suppression of the longevity phenotype (~30% lifespan extension in N2 vs ~12% extension in the *sek-1* mutant).

Response: Thank you for your kind comment and suggestion. We took this

Reviewer’s suggestion and carefully interpreted the conclusion that OP50+Probio-M9

was completely unable to extend the lifespans of *nsy-1 (ag3)*, *sek-1 (km4)* or *pmk-1 (km25)* mutant worms. Please see line 207-210 of the Article.

Interestingly, we found that *nsy-1 (ag3)* and *pmk-1 (km25)* mutation almost completely suppressed the lifespan extension associated with Probio-M9 feeding, *sek-1 (km4)* mutation strongly but incompletely suppressed the longevity (~30% lifespan extension in N2 vs ~12% lifespan extension in the *sek-1* mutant).

5. Page 12: “OP50+Probio-M9 was unable to extend the lifespan of the *isp-1 (qm150)* or *nuo-6 (qm200)* mutants, suggesting that the ETC in mitochondria is indispensable for the lifespan-extending property of Probio-M9.” The authors need to interpret these data carefully. Firstly, both *isp-1 (qm150)* or *nuo-6 (qm200)* are not null mutants, but instead are hypomorphs due to substitution of a single amino acid in these proteins. Hence, it cannot be concluded that ETC proteins are indispensable for the lifespan extension phenotype of Probio-M9. Secondly, both *isp-1 (qm150)* or *nuo-6 (qm200)* are extremely long-lived compared to wild-type N2 animals. Hence, the most likely interpretation of the data reported in Figs 6d, e is that the lifespan extending effects of Probio-M9 converge with the pro-longevity effects of reduced mitochondrial function in worms. The authors should also change the claim that “Probio-M9 extends lifespan likely through interaction with the mitochondrial ETC complexes”.

Response: Thank you for your kind comment and suggestion. We agree with the Reviewer’s criticism about the interpretation of the fact that OP50+Probio-M9 was unable to extend the lifespan of the *isp-1 (qm150)* or *nuo-6 (qm200)* mutants. We have

modified it. Please see line 274-278 of the Article.

Again, OP50+Probio-M9 was unable to extend the lifespan of the *isp-1* (*qm150*) or *nuo-6* (*qm200*) mutants (Fig. 6d, e and Supplementary Table 7), suggesting that the lifespan extending effects of Probio-M9 converge with the pro-longevity effects of reduced mitochondrial function in *C. elegans*.

We also modified the claim that “Probio-M9 extends lifespan likely through interacting with the mitochondrial ETC complexes”. Please see line 278-280 of the Article.

These results showed that Probio-M9 extends its host’s lifespan by enhancing the UPR^{mt} and thus maintaining mitochondrial homeostasis in intestinal cells.

Minor comments:

a) Abstract: “Probio-M9 protects and repairs damaged the mitochondria” – replace with ‘Probio-M9 protects and repairs damaged mitochondria’.

Response: Thank you for your kind comment. We have replaced the content “Probio-M9 protects and repairs damaged the mitochondria” with “Probio-M9 protects and repairs damaged mitochondria”. Please see line 37-38 of the Article. Probio-M9 protects and repairs damaged mitochondria by activating mitochondrial unfolded protein response.

b) The authors should cite the study (PMID: 35127565) that has previously suggested a role of the *pmk-1* (p38 MAPK) signaling pathway in extending lifespan of *C.*

C. elegans fed with the GG strain of *Lacticaseibacillus rhamnosus*.

Response: Thank you for your kind suggestion. We have added the citation of this paper (PMID: 35127565) in the Discussion section, please see line 400-405 (Reference 62) of the Article.

Our results showed that OP50+Probio-M9 was unable to extend the lifespan of loss-of-function *pmk-1* mutant, indicating that the p38 cascade might be involved in the longevity related to Probio-M9 feeding, consistent with a previous report of *Lacticaseibacillus rhamnosus* GG extending the lifespan of *C. elegans* by activating the *pmk-1* (p38 MAPK) signaling pathway (Reference 62).

62 Yun, B. *et al.* Probiotic *Lacticaseibacillus rhamnosus* GG increased longevity and resistance against foodborne pathogens in *Caenorhabditis elegans* by regulating MicroRNA miR-34. *Frontiers In Cellular And Infection Microbiology* **11**, 819328 (2021).

c) Page 4: “*daf-2* is a key upstream component of IIS, regulating various physiological processes, such as aging and adult lifespan in *C. elegans*” – Here the authors should also cite the first study that showed the longevity phenotype of *daf-2* mutants (PMID: 8247153). In addition, they should describe that the lifespan extending effects of *daf-2* are mediated via the *daf-16*/FoxO transcription factor in reduced insulin signaling mutants such as *daf-2 (e1370)*.

Response: Thank you for your kind suggestion. We have cited the paper (PMID: 8247153) in the Introduction section, which suggested that the lifespan extending

effects of *daf-2* are mediated via the *daf-16*/FOXO transcription factor in reduced insulin signaling mutants such as *daf-2 (e1370)*, please see line 66-70 (Reference 20) of the Article.

The sole insulin/IGF-1 receptor encoded by the gerontogene *daf-2* is a key upstream component of IIS, regulating various physiological processes, such as aging and adult lifespan in *C. elegans* (Reference 20). The lifespan extending effects of *daf-2* are mediated via the *daf-16*/FOXO transcription factor by reducing insulin signaling mutants such as *daf-2 (e1370)* (Reference 20).

20 Kenyon, C., Chang, J., Gensch, E., Rudner, A. & Tabtiang, R. A *C. elegans* mutant that lives twice as long as wild type. *Nature* **366**, 461-464 (1993).

d) Page 4: “SKN-1, a stress-responsive nuclear respiratory factor” – replace ‘respiratory factor’ with ‘transcription factor’.

Response: Thank you for your kind comment. We have replaced the content “SKN-1, a stress-responsive nuclear respiratory factor” with “SKN-1, a stress-responsive nuclear transcription factor”. Please see line 70-72 of the Article.

SKN-1, a stress-responsive nuclear transcription factor, contributes to the reduction of IIS-associated longevity (Reference 22).

22 Tullet *et al.* Direct inhibition of the longevity-promoting factor SKN-1 by insulin-like signaling in *C. elegans*. *Cell* **132**, 1025-1038 (2008).

e) Page 7: The authors claim that Probio-M9 extends lifespan in a dietary restriction

(DR)-independent manner. They make this claim based on findings that feeding worms with Probio-M9 does not produce any significant effect on developmental rate, brood size or body length. To conclusively demonstrate that Probio-M9 does not extend lifespan via dietary restriction, the authors should also discuss the results for the very well-studied genetic model of DR, the *eat-2 (ad1116)* strain, in this section (Supplementary Fig. 2c).

Response: Thank you for your kind comment and suggestion. We have added the results for the very well-studied genetic model of DR, the *eat-2 (ad1116)* and *aak-2 (ok524)* mutants, in the Results section to verify Probio-M9 extending lifespan in a dietary restriction (DR)-independent manner (adjusted Supplementary Fig. 2c to Fig. 2e, f). Please see line 133-136 and Fig. 2e, f of the Article.

In addition, we investigated the effect of DR signaling pathway in lifespan extension of *C. elegans*. We found that OP50+Probio-M9 still extended the lifespans of *eat-2 (ad1116)* and *aak-2 (ok524)* mutant worms (Fig. 2e, f and Supplementary Table 2).

f) In Supplementary Figs 1b and 1d, the y-axes represent percentage. Hence, error bars above 100% are misleading. In addition, t-tests are not appropriate for

comparison of proportions. The authors should use Chi-squared test for proportions instead.

Response: Thank you for your kind comment and suggestion. We have modified the y-axes representing percentage in Supplementary Figs 1b and 1d and reanalyzed the statistical proportions using the Chi-squared test. Please see Supplementary Figs 1b and 1d. Both the figures and figure legends are now updated.

g) In the Supplementary Tables for lifespan data, the authors should also list the total number of animals for each condition/genotype and how many of those were censored during the lifespan analysis.

Response: Thank you for your kind comment and suggestion. We have added the total number of animals for each condition/genotype (Total N) and the number of animals censored (censor N) during the lifespan analysis, please see the Supplementary Tables.

h) In the Supplementary Tables for lifespan data, the actual *p*-value should be listed

instead of ' $p > 0.05$ '. This is because in Supplementary Table 3, lifespan extension up to 13% has been reported as non-significant, while in Supplementary Table 2, a 7% lifespan extension is reported as statistically significant. The actual p -value for these conditions needs to be reported to know whether the effects reported as 'non-significant' were approaching significance ($p < 0.1$).

Response: Thank you for your kind suggestion. We agree with the Reviewer's criticism about listing the actual p -value in the Supplementary Tables. We have modified the actual p -value for lifespan data, please see the Supplementary Tables.

i) In Fig. 1b, 'OP50+Probio-M9' should be written as 'OP50+4 Probio-M9'.

Otherwise, the reader might assume that these two bacteria were present in a 1:1 ratio (based on the other conditions used in this experiment).

Response: Thank you for your kind comment and suggestion. We do understand and agree with the concern of this Reviewer, and we changed the proportion of Probio-M9 in the bacterial mixture in a 10-fold gradient in this experiment. In response, we have modified 'OP50+Probio-M9' to 'OP50+4 Probio-M9', and correspondingly modified 'OP50+0.1 Probio-M9' to 'OP50+0.4 Probio-M9', modified 'OP50+0.01 Probio-M9' to 'OP50+0.04 Probio-M9', and modified 'OP50+10 Probio-M9' to 'OP50+40 Probio-M9', please see the Fig. 1b. Both the figures and figure legends are now updated.

j) Fig. 2f: What statistical tests were used for these data? Chi-squared tests for proportions should be used and the p -values need to be corrected for multiple comparisons.

Response: Thank you for your kind comment and suggestion. The statistics tests used were Chi-squared test, and Chi-squared test coupled with multiple comparisons of corrected p -values was used for normal locomotor significance analysis (adjusted Fig. 2f to Fig. 2h). The statistics analysis method is described in details in the Methods section, please see line 658-663 of the Article. Both the figures and figure legends are now updated.

For statistical data, Chi-squared test coupled with multiple comparisons were used for

significance analysis. In all cases, $p < 0.05$ was considered significant. In the figures, the asterisk indicates the statistical significance of the Chi-squared test analysis as compared to control.

k) Figs 2h, 2i: Were the p -values for the t-tests corrected for multiple comparisons?

Response: Thank you for your kind comment and suggestion. Yes, the p -values for the Student's t test were corrected for multiple comparisons, and the statistics analysis method is described in details in the Methods section, please see line 658-663 of the Article.

For statistical data, Student's t test coupled with multiple comparisons were used for significance analysis. In all cases, $p < 0.05$ was considered significant. In the figures, the asterisk indicates the statistical significance of the Student's t test analysis as

compared to control.

In addition, we adjusted Figs 2h, 2i to Fig. 2j, 2k. Both the figures and figure legends are now updated.

1) Page 9: The authors make the claim that “We also found that worms cultured from eggs to adulthood (day 2 or day 4 after L4 stage) on OP50+Probio-M9 had a significantly longer lifespan extension than those cultured from eggs to the L4 stage on OP50+Probio-M9”. However, they did not perform the statistical analysis for this comparison. Since they performed all these lifespan experiments in triplicates, they can perform statistical tests on the lifespan extension values for the three types of treatments. One-way ANOVA followed by post-hoc tests would be ideal for this comparison.

Response: Thank you for your kind comment and suggestion. We have performed statistical tests on the lifespan extension values for the three types of treatments, please see line 174-177 and Fig. 3f of the Article. Both the figures and figure legends are now updated.

We also found that worms cultured from eggs to adulthood (day 4 after L4) on OP50+Probio-M9 had a significantly longer lifespan extension than those cultured from eggs to the L4 stage on OP50+Probio-M9 (Fig. 3f and Supplementary Table 3).

f

m) Fig. 7c – The color of this heatmap should be changed to red-blue format to make it comprehensible for colorblind individuals.

Response: We thank you for your suggestion of changing the color format. We have modified the color of the heatmap by changing red-green to red-blue in the Fig. 7c.

Please see the Fig. 7c of the Article.

C

Reviewer #2 (Remarks to the Author):

We thank this reviewer for the overall positive assessment of the study.

In this manuscript, Zhang et al investigate the effect of *L. rhamnosus* Probio-M9 supplementation on lifespan extension in a model nematode, *C. elegans*. They also elucidated molecular mechanism and metabolic changes underlying the effect of probiotic bacteria on the lifespan extension.

So far, the probiotic bacteria-mediated lifespan extension is already known in the previous research, but authors elucidate detailed molecular pathway of the probiotic's effects such as p38 and *daf-2* signaling, mitochondrial unfolded protein response. One of main novel point of the present investigation is the metabolite analysis. Research topic of present study seems proper for Communications Biology. Authors careful design and execute experiment to support their hypothesis. They provide experimental data using various mutant *C. elegans* strain, and provide some novel knowledge to researchers who develop nutraceuticals for antiaging purpose. However, authors need to consider and answer the following points.

Thank you very much for your positive comments, which have greatly improved the quality of our manuscript. We will reply to your comments one by one.

1) Authors analyze metabolic changes of *C. elegans* after the supplementation of probiotic bacteria by using UPLC analysis. But the detailed methods how authors identify and quantify each metabolite, therefore, authors need to describe the

metabolomics analysis method more in detail in the Methods section.

Response: Thank you for your kind comment and suggestion. We have added the metabolomics analysis method in details in the Methods section. Please see line 630-653 of the Article.

Software Analyst 1.6.3 was used to process the mass spectrum data. The repeatability of metabolite extraction and detection were estimated using TIC and multi peak multiple reaction monitoring (MRM). According to home-made metadata database (MWDB) and other databases, qualitative analysis of information and secondary general data was carried out based on retention time (RT) and mass-to-charge ratio. The metabolite structure analysis referred to several public mass spectrometry databases, including Human Metabolome Database (HMDB, <http://www.hmdb.ca/>), Metlin (<http://metlin.scripps.edu/index.php>), massbank (<http://www.massbank.jp/>) and knapsack (<http://kanaya.naist.jp/knapsack/>).

Ten microliters per sample was used for quality control (QC) of metabolomics analysis. When running sample sets on column, one QC sample was injected after 10 samples. Metabolite quantification was analyzed using MRM of triple quadrupole mass spectrometry. The mass spectrum file under the sample machine was opened with multi-quantitative software, and the chromatographic peaks were integrated and calibrated. The peak area of each chromatographic peak represented the relative content of the corresponding substance. The integral data of chromatographic peak area was extracted and saved, and the positive and negative ions of metabolites were removed using self-built software package.

The coefficient of variation (CV) values of metabolites in QC samples were calculated, and metabolites with CV values greater than 0.5 were removed. When metabolites were detected in both positive and negative ionization modes, those with larger CVs were removed in either positive or negative mode. Orthogonal partial least squares discriminant analysis (OPLS-DA) model was used to identify differences in metabolic profiles between groups.

2) Author just list up differently present metabolite and analyze the KEGG pathway. However, these seem not enough for the present study. Author need to execute some supplementation experiment of the most important metabolites and prove the functionality or causality of metabolite related to the lifespan modulation effects of probiotics. Author need to discuss meaning of important metabolites related to aging and anti-aging, based on the previous research.

Response: Thank you for your kind comment, we agree with the Reviewer's suggestion of performing some supplementation experiments on the most important metabolites. The Probio-M9 used in our study has previously been survived well in high bile salts or at a low pH *in vitro* (probiotics screen, Reference 44) and inhibited the formation of colorectal tumors in mice (animal model, Reference 45). Metabolomics analysis in this study found that differentially expressed metabolites were mainly enriched in pathways involving amino acid metabolism, galactose metabolism, sphingolipid metabolism, fatty acid biosynthesis and cAMP signaling. It has been reported that these bacterial metabolites play crucial roles in host aging

(Reference 57,78-82). Therefore, we reasoned that the key metabolites secreted by Probio-M9 might contribute to delaying aging in the host or age-related chronic diseases. Due to the imperfection of the extraction method for the key metabolites of Probio-M9, various possible methods are being tried depending on the extraction tool and product concentration. Next, we would continue to optimize the extraction method of the most key metabolites in Probio-M9 and execute some lifespan verification experiments and prove the health-promoting effects and longevity of probiotic metabolite on host.

Thank you again for your kind comment. We have added the meaning of key metabolites related to aging and anti-aging in the Discussion section, based on the previous research. Please see line 458-487 of the Article.

Amino acids affect the longevity of the host, tryptophan and proline contribute to delaying the aging of *C. elegans*, while phenylalanine accelerates aging (Reference 78); aspartate, glutamate and methionine extend the lifespan of yeast (Reference 79).

Our results suggested that the significant enrichment of bacterial tryptophan, aspartate and glutamate metabolites could be related with the extension of the lifespan of *C. elegans* by Probio-M9. Bacterial glucose metabolites, such as glucose, galactose and lactose, may also be associated with host longevity participate in anti-aging activities (Reference 57). Our metabolomics data indicated that the significant enrichment of bacterial galactose metabolites may contribute to the extension of the lifespan of *C. elegans* by Probio-M9. Consistent with our findings, Zanni et al. found that galactose secreted by *Lactobacillus delbrueckii* subsp. *bulgaricus* might be associated with its

beneficial effects on *C. elegans* (Reference 80). The accumulation of bacterial metabolites such as sphingolipid and ceramide are closely related to aging and age-related diseases (Reference 81,82). It has been shown that sphingolipid metabolism might be vital for enhancing locomotor ability (Reference 83) and improvement of the health status of *C. elegans*, which were partially dependent on the insulin receptor *daf-2* signaling pathway (Reference 84). Goya et al. observed that *Bacillus subtilis* mediated host sphingolipid metabolism and inhibited *a-syn* aggregation in *C. elegans* (Reference 85). In light of our results, we revealed that Probio-M9 might regulate sphingolipid metabolism through nutrient sensing to extend lifespan of *C. elegans*. Besides, our data showed that 2-aminoethanesulfonic acid derived from taurine metabolism was significantly accumulated upon OP50+Probio-M9 feeding. Taurine is of great significance in the scavenging of lipid metabolism, immune function, and other physiological functions, making it an important promoter of anti-inflammatory activity (Reference 86,87). However, such inference needs further experimental verification. It will be of interest to extract the most key metabolites in Probio-M9 and perform further lifespan verification experiments to demonstrate the health-promoting and longevity extension effects of probiotic metabolites on host. It is necessary to identify the major anti-aging components of probiotics and explore signaling pathways that probiotic utilizes to delay host aging.

44 Liu, W. *et al.* Characterization of potentially probiotic lactic acid bacteria and bifidobacteria isolated from human colostrum. *Journal of Dairy Science* **103**,

- 4013-4025 (2020).
- 45 Xu *et al.* Inhibitory effects of breast milk-derived *Lactobacillus rhamnosus* Probio-M9 on colitis-associated carcinogenesis by restoration of the gut microbiota in a mouse model. *Nutrients* **13**, 1143 (2021).
- 57 Brokate-Llanos *et al.* *Escherichia coli* carbon source metabolism affects longevity of its predator *Caenorhabditis elegans*. *Mechanisms of Ageing & Development* **141-142**, 22-25 (2014).
- 78 Edwards, C. *et al.* Mechanisms of amino acid-mediated lifespan extension in *Caenorhabditis elegans*. *BMC Genetics*, *16*,1(2015-02-03) **16**, 8 (2015).
- 79 Wu, Z., Song, L., Shao, Q. L. & Huang, D. Independent and additive effects of glutamic acid and methionine on yeast longevity. *PloS One* **8**, e79319 (2013).
- 80 Zanni, E. *et al.* Combination of metabolomic and proteomic analysis revealed different features among *Lactobacillus delbrueckii* Subspecies *bulgaricus* and *lactis* strains while *in vivo* testing in the model organism *Caenorhabditis elegans* highlighted probiotic properties. *Frontiers In Microbiology* **8**, 1206 (2017).
- 81 Perez, G. I. A central role for ceramide in the age-related acceleration of apoptosis in the female germline. *Faseb Journal Official Publication of the Federation of American Societies for Experimental Biology* **19**, 860 (2005).
- 82 Venable *et al.* Shift in sphingolipid metabolism leads to an accumulation of ceramide in senescence. *Mechanisms of Ageing & Development* **127**, 473-480 (2006).
- 83 Chan, J. P., Brown, J., Hark, B., Nolan, A. & Staab, T. A. Loss of sphingosine

kinase alters life history traits and locomotor function in *Caenorhabditis elegans*.

Frontiers in Genetics **8**, 132 (2017).

- 84 Mai-Britt *et al.* Functional loss of two ceramide synthases elicits autophagy-dependent lifespan extension in *C. elegans*. *PloS One* **8**, e70087 (2013).
- 85 Goya *et al.* Probiotic *Bacillus subtilis* protects against α -synuclein aggregation in *C. elegans*. *Cell Reports* **30**, 367-380 e367 (2020).
- 86 Aydn *et al.* Carnosine and taurine treatments diminished brain oxidative stress and apoptosis in D-galactose aging model. *Metabolic Brain Disease* **31**, 337-345 (2016).
- 87 Suliman *et al.* Accumulation of taurine in patients with renal failure. *Nephrology, dialysis, transplantation : official publication of the European Dialysis and Transplant Association - European Renal Association*, 528-529 (2002).

3) In Abstract, author need to precisely describe change of metabolism is originated from *C. elegans* or probiotic bacteria itself. Last sentence of Abstract should be revised.

Response: Thank you for your meaningful comment. It makes sense that the change of metabolism is originated from *C. elegans* or probiotic bacteria itself, which is the key to this study. We have added the description stating change in metabolism is originated from probiotic bacteria itself, please see line 39-40 of the Article.

The significant increase of amino acids, sphingolipid, galactose and fatty acids in

bacterial metabolites might be involved in extending the lifespan of *C. elegans*.

Thank you again for your kind comment. We have modified last sentence of Abstract, please see line 40-43 of the Article.

We reveal that Probio-M9 as a dietary supplementation had the potential to delay aging in *C. elegans* and also provide new methods and insights for further analyzing probiotics in improving host health and delaying the occurrence of age-related chronic diseases.

4) First paragraph of Result section, 1:4 ratio of bacterial mixture=> 2.82 or 2.72×10^9 CFU/ml of Probio-M9?

Response: Thank you for your kind comment. In this study, viable plate count of Probio-M9 was 2.82×10^9 CFU/mL, and we used an approximately 1:4 ratio of *E. coli* OP50 to Probio-M9. We have modified the description, please see line 99-102 of the Article.

In this study, we used an approximately 1:4 ratio of *E. coli* OP50 to Probio-M9 (viable plate count: 6.8×10^8 CFU/mL and 2.82×10^9 CFU/mL, respectively) as the experimental group (OP50+Probio-M9), and *E. coli* OP50 alone as the control group (OP50).

REVIEWERS' COMMENTS:

Reviewer #1 (Remarks to the Author):

The authors have satisfactorily addressed all my comments and the revised manuscript is much improved. Some minor comments are listed below.

(a) Line 69: This should be rephrased as 'The lifespan extending effects of daf-2 are mediated via the daf-16/FOXO transcription factor in reduced insulin signaling mutants such as daf-2 (e1370)'.

(b) Line 96: Replace 'genu' to 'genus' (or ideally the plural form 'genera').

(c) Figures 3b, 3d and 3e look almost identical. Some label should be added to the figures to demonstrate the differences between these panels. This information is provided in the figure legend but adding short labels (such as 'Switch to OP50 on day 2') directly on the figure will be helpful to the readers.

13 September 2022

Zhang et al. *Communications Biology* Tracking #:

COMMSBIO-22-0589-T

Response to reviewer comments

We are pleased to hear from you. We appreciate your interest, time and labor on our manuscript. We are most grateful for the positive and constructive comments provided by reviewer and we revised manuscript accordingly in order to address your concerns.

Our point-by-point responses to the reviewer comments are provided below. The reviewers' comments are provided in black Roman typeface and our responses are written in red Roman typeface after each comment.

REVIEWERS' COMMENTS:

Reviewer #1 (Remarks to the Author):

Thank you for your kind remarks and positive advice.

The authors have satisfactorily addressed all my comments and the revised manuscript is much improved. Some minor comments are listed below.

Thank you for your kind remarks and helpful advice, which have greatly improved the quality of our manuscript. We will respond to your comments one by one. Please check the specific reply below.

(a) Line 69: This should be rephrased as ‘The lifespan extending effects of *daf-2* are mediated via the *daf-16*/FOXO transcription factor in reduced insulin signaling mutants such as *daf-2 (e1370)*’.

Response: Thank you for your kind comment. We have replaced the content “The lifespan extending effects of *daf-2* are mediated via the *daf-16*/FOXO transcription factor by reducing insulin signaling mutants such as *daf-2 (e1370)*” with “The lifespan extending effects of *daf-2* are mediated via the *daf-16*/FOXO transcription factor in reduced insulin signaling mutants such as *daf-2 (e1370)*”. Please see line 69-71 of the Article.

The lifespan extending effects of *daf-2* are mediated via the *daf-16*/FOXO transcription factor in reduced insulin signaling mutants such as *daf-2 (e1370)*.

(b) Line 96: Replace ‘genu’ to ‘genus’ (or ideally the plural form ‘genera’).

Response: Thank you for your kind comment. We have replaced the content “genu” with “genera”. Please see line 97 of the Article.

Probio-M9 and *E. coli* OP50 belong to different bacterial genera, it has been reported that nematodes exhibited a preference when the normal food *Escherichia coli* OP50 (*E. coli* OP50) is replaced with other bacteria.

(c) Figures 3b, 3d and 3e look almost identical. Some label should be added to the figures to demonstrate the differences between these panels. This information is provided in the figure legend but adding short labels (such as ‘Switch to OP50 on day 2’) directly on the figure will be helpful to the readers.

Response: Thank you for your kind comment and suggestion. We have added the label to the figures to demonstrate the differences between these panels. Please see Fig. 3b, 3d and 3e of Figure 3.